

# Assessment of uncertainties in soil erosion and sediment yield estiamtes at ungauged basins: an application to the Garra River basin, India

Somil Swarnkar[1], Anshu Malini[2], Shivam Tripathi[3] and Rajiv Sinha[1]

[1]Department of Earth Sciences, Indian Institute of Technology Kanpur, Kanpur 208016, India
[2]Department of Irrigation, Secunderabad 500003, India
[3]Department of Civil Engineering, Indian Institute of Technology Kanpur, Kanpur 208016, India

*Correspondence to:* Rajiv Sinha (rsinha@iitk.ac.in)

**Abstract.** High soil erosion and excessive sediment load are serious problems in several Himalayan River basins. To apply mitigation procedures, precise estimation of soil erosion and sediment yield with associated uncertainties are urgently needed. Here, Revised Universal Soil Loss Equation (RUSLE) and Sediment Delivery Ratio (SDR) equations are used to estimate the spatial pattern of soil erosion (SE) and sediment yield (SY) in the Garra River basin, a small Himalayan tributary of River Ganga. A methodology is proposed for quantifying and propagating uncertainties in SE, SDR and SY estimates. Expressions for uncertainty propagation are derived by first-order uncertainty analysis, making the method viable even for large river basins. The methodology is applied to investigate the relative importance of different RUSLE factors in estimating the magnitude and uncertainties of SE over two distinct morpho-climatic regimes of the Garra River basin, namely, upper mountainous region & lower alluvial plains. The results suggest that average SE in the basin falls in very high category (20.4 ± 4.1 t/ha/y) with higher values in the upper mountainous region (84.4 ± 13.9 t/ha/y) than in the lower alluvial plains (17.7 ± 3.6 t/ha/y). Furthermore, the topographic steepness (LS) and crop practice (CP) factors exhibit higher uncertainties than other RUSLE factors. The annual average SY is estimated at two locations in the basin - Nanak Sagar dam (NSD) for the period 1962-2008 and Husepur gauging station (HGS) for 1987-2002. The SY at NSD and HGS are estimated to be $8.0 \pm 1.4 \times 10^5$ t/y and $7.9 \pm 1.7 \times 10^6$ t/y, respectively, and the estimated 90% confidence interval contains the observed values $6.4 \times 10^5$ t/y and $7.2 \times 10^6$ t/y. The study demonstrated the usefulness of the proposed methodology for quantifying uncertainty in SE and SY estimates at ungauged basins.

## 1. Introduction

Soil erosion is a serious problem, which not only causes land degradation and loss of agricultural productivity but also alters geomorphic processes and sediment fluxes in a river basin. Estimation of soil erosion (SE) and sediment yield (SY) of a river basin are therefore essential for agricultural planning and river management. SE and SY can be estimated by either empirical models that are developed solely based an experimental studies (e.g. USLE, RUSLE and MUSLE) or process based models that are based on parameterization of physical process (e.g. WEPP, CREAMS and



AGNPS). While the process based models may be more reliable and appealing, the empirical models are popular because they can be applied on basins with no or limited data (Merritt et al., 2003).

The estimates of SE and SY alone are not sufficient for effectively addressing soil erosion problem in a river basin. One needs to quantify uncertainties in those estimates as well (Beven and Brazier, 2011). These uncertainties can stem from

uncertainties in data (measurement errors, coarse spatial and temporal resolution, missing values), uncertainties in model (parameter uncertainty, structural uncertainty, algorithmic or numerical uncertainty), and stochastic nature of the soil erosion process. Since quantification of all the sources of uncertainties is difficult, studies make assumptions about their relative importance and mutual independence. Nevertheless, to the best of authors' knowledge no unified approach is available to quantify uncertainty in SE and SY for ungauged basins. *The main objective of this work is to present a*

*methodology to quantify uncertainties in SE and SY for ungauged basins using commonly used models and easily accessible datasets*. The following paragraphs briefly review the literature on the uncertainty estimates of SE and SY, and highlight the existing gaps.

Arguably, the most popular empirical model for estimating SE is USLE and its variants such as RUSLE and MUSLE. Variations of USLE are also used in distributed hydrological models like EUTROMOD, SWAT and SWRRB. The

USLE estimates sheet and rill erosion, but does not account for gully or channel erosion in a basin. Developed in 1960s with more than 10,000 plot-yr data from USA, the method was designed for estimating long-term SE at a plot-scale, but it is now frequently used for estimating erosion at a basin-scale, albeit with some modifications. This study uses RUSLE model that estimates SE by considering five factors, namely, rainfall and runoff erosivity (R factor), soil erodibility (K factor), topography (LS factor) and crop management practices (CP factor).

The studies on uncertainty analysis of RUSLE equation can be divided into three types–

(a)  Studies that have quantified uncertainties in individual factors of the RUSLE equation. For example, Wang et al. (2002b) quantified spatial uncertainty in R factor by using geostatistics. Catari et al. (2011) assessed uncertainty in R factor by comparing traditional methods with at-site measurements. Torri et al. (1997), Wang et al. (2001) and Parysow et al. (2003) investigated uncertainty in K factor by using geostatistical methods. Gertner et al. (2002),

25       Wang et al. (2002a), Wu et al. (2005) and Mondal et al. (2016) estimated uncertainty in LS factor based on at-site measurements and cell variation in digital elevation model (DEM).

(b)  Studies that have used backward or inverse uncertainty propagation in which modeled and observed values of SE are compared to evaluate model biases, check model's suitability for a basin, and estimate model parameters (Risse et al., 1993; Falk et al., 2010 and Carmona et al., 2017). The backward uncertainty analysis requires observed values

30       of SE, and hence not applicable for ungauged basins.



(c)     Studies that have used forward uncertainty propagation in which uncertainties present in input data and/or model are propagated to quantify uncertainties in the SE estimates. Three such studies that are relevant for this work are summarized below –

(i)     Hession et al. (1996) presented a two-phase Monte Carlo methodology for forward propagation of uncertainty and demonstrated its application at an experiential plot in Oklahoma, USA. They divided uncertainty in USLE factors into knowledge uncertainty and natural stochastic variability, and argued that the two types of uncertainties should be analyzed separately to draw useful conclusions. They considered knowledge uncertainty for R factor, stochastic variability for K and C factors, and treated LS and P factors as constants. They also studied the effect of spatial discretization in models and the assumption of independence of parameters on uncertainty quantification. However, since information on dependence of USLE factors was not available, they assumed different levels of correlation among them.

(ii)    Biesemans et al. (2000) applied Monte Carlo error propagation technique for estimating uncertainty in SE and SY (referred to as off-site sediment accumulation) at a watershed in Belgium.  Elevation data were assumed to have auto-correlated errors, which were modeled using fractional Gaussian noise to estimate uncertainty in LS factor. Soil texture and its organic content were measured at 153 locations in the watershed to estimate K factor. The K factor so obtained was interpolated using Kriging. The variance of the Kriging surface was taken as a measure of K factor uncertainty. The C factor was assumed to have uniform distribution with minimum and maximum values estimated based on USLE table and by appropriately weighing C factor for each crop by the erosivity value in its growing season. R and P factors were assumed constant. The result showed that the observed value of SE lies within one standard error (68%) of the estimated mean value, prompting authors to conclude that RUSLE is a suitable model for their study watershed and that the RUSLE model should use probability distribution of input factors rather than their fixed values.

(iii)   Tetzlaff and Wendlend (2012) and Tetzalf et al. (2013) performed forward uncertainty analysis on ABAG model (an adaptation of USLE to German conditions) by using Gaussian error propagation and Monte Carlo simulations.  ABAG was a part of MEPhos model that was applied to determine SY for the state of Hesse in Germany ($21,115$ km$^2$). However, because of high computational cost, the uncertainty analysis for SE using ABAG was performed for a relative small catchment of River Gersprenz ($485$ km$^2$). The uncertainty in LS factor was estimated as standard deviation of 1,000 LS factors derived from 1,000 simulated DEM surfaces obtained by adding random Gaussian error to the original DEM. Uncertainties for the other USLE factors were assumed (R- 10%, C – 23%, and K-10%) based on auxiliary information, and P factor was treated as constant. The authors calculated that the uncertainty in USLE factors resulted in 34% uncertainty in the mean annual soil loss estimates.

Not all of the sediment eroded in a basin is delivered out of it; a significant portion of the eroded material gets deposited at intermediate locations. Sediment yield (SY) denotes the total sediment outflow from a basin over a specified





duration. It is usually measured by either streamflow sediment sampling or reservoir sedimentation survey. By definition, SY includes both bed load and suspended load. However, since streamflow sediment sampling is often restricted to suspended load, the SY estimates from streamflow sampling are usually adjusted upward by some empirical procedure (e.g. Table 3.2 of Vanoni, 1975).

Reviews on SY modeling suggest that unlike SE modeling, no universal relationships are available that can be applied to every situation, rather region specific relationship is considered to be the best method for predicting SY (Ludwig and Probst, 1998; De Vente et al., 2011). The most common approach to predict SY is to estimate it as a product of gross SE and sediment delivery ratio (SDR; Walling, 1983; Richards, 1993), where SDR is defined as the ratio of SY at a prediction location to the gross or total SE of a basin whose outlet is the prediction location. Precise estimate of SDR is
not available, but it is primarily related to the drainage basin area (USDA, 1972; De Vente et al., 2007). According to Boyce (1975), SDR generally decreases with increasing basin area because with increase in basin size mean slope decreases and sediment storage locations between source areas & the basin outlet increases. The most favored method for long-term SY estimation is USLE-SDR method where gross SE is estimated by the USLE model (e.g., Ebisemiju, 1990; Walling, 1993; Van et al., 2001; Amore et al., 2004; Bhattarai and Dutta, 2007; Boomer et al., 2008). This study
uses RUSLE-SDR model for predicting SY, in which SDR is obtained as a function of basin area based on the equation developed for north Indian River basins by Sharda and Ojasvi (2016).

Only a few studies have reported uncertainty in SY estimates by using USLE-SDR approach. Ferro & Porto (2000) and Stefano & Ferro (2007) quantified uncertainty in the estimates of SY for river basins in Italy using a USLE-SDR based model termed sediment delivery distributed (SEDD) model. The model could predict SY at event and annual scales, but
requires observed data for its calibration and hence not suitable for ungauged basins. Catari (2010) used RUSLE-SDR model to investigate uncertianty in the estiamtes of SY for the Upper Llobregat River basin in Spain. The uncertainites in individual RUSLE factors and SDR are first quantified and then added in quadrature to estimate uncertainty in SY.The LS and C factors were found to have the major influence on SY uncertainty.

The literature review on SE and SY estimation described in the foregoing paragraphs suggest that – (i) Very few studies
have computed uncertainties in SE and SY for ungauged basins. Most of the existing studies are either restricted to plot-scale or are carried out for basins with measured data; (ii) While the importance of sediment erosion in Himalyan basins is well known (Galy and France-Lanord, 2001; Rahaman et al. 2009) , no studies are available that have quantifed uncertainties in SE and SY estiamtes for these basins; (iii) Presence of storage strucures, like dams and reservoirs, complicates the estimation of SY downstream of the structure. Though some simplified methods exist to account for
control structures in SY estimation (Sharda and Ojasvi, 2016), their effects on the uncertainty quantification has not been explored.



The aim of this study is to develop a methodology for determining uncertainties in SE and SY estimates of ungauged basins. The Garra River, a Himalayan tributary of River Ganga, was selected for demonstration of the developed methodology and for investigating the role of uncertainties in input parameters on uncertainties in the estimates of SE and SY. The specific objectives of this study are:

(i)     To estimate spatially distributed SE for the Garra River basin using the RUSLE model.

(ii)    To quantify uncertainty in SE estimate by accounting for uncertainties in different RUSLE factors.

(iii)   To study the relative importance of different RUSLE factors in governing erosion and its uncertainty over mountainous and alluvial plain regions of the basin.

(iv)    To estimate SDR for the Garra River basin and its uncertainty.

(v)     To evaluate SY for the basin by combining SE and SDR estimates.

Remainder of the paper is organized into four sections. Section 2 describes the study area and data used. Section 3 presents the methodology for estimating SE, SDR and SY, and for quantifying uncertainties in these estimates. Section 4 describes the results obtained, and Section 5 lists the limitations of the proposed methodology. Finally, Section 6 summarizes the major finding of this work and presents a set of concluding remarks.

## 2.   Study Area & Data Used

The study basin is the Garra/Deoha River, a Himalayan tributary of River Ganga. This river originates near Haldwani in Uttarakhand from a lake (29°12'16" N, 79°45' 30" E) fed by glacier melt (Roy and Sinha, 2007), and meets River Ganga near Bilgram and Kannauj in Uttar Pradesh (27°08'30" N, 79°56'40" E). The study basin is located between 27°09' N to 29°18' N latitude and 79°38' E to 80°09' E longitude, covering a total area of around 7,000 km$^2$ (Fig. 1). The Garra basin has two distinct morpho-climatic regimes, upper mountainous region (part of Himalayan foothill) and lower alluvial plains (part of upper Indo-Gangetic plains). The upper mountainous region of the basin has a high average annual rainfall of 1500 mm, and the lower alluvial part has a comparatively lower average annual rainfall of 1050 mm (Fig. 2d).

The only gauging station in the basin is at Husepur (27°16'30" N, 79°57'0.64" E) near Hardoi, Uttar Pradesh, which was operated by Central Water Commission (CWC) from 1987-2002. The discharge and suspended sediment load records for 16 years (1987 – 2002) are available at this station. Topography, land use land cover (LULC), soil and rainfall datasets are obtained from different open data sources listed in Table 1. The table also provides the spatial and temporal resolutions, and temporal extent of these datasets.

The Garra River has a major intervention in the form of Nanak Sagar reservoir (28°57' 10" N, 79°50'30" E; capacity 210 Mm$^3$) created by a dam of the same name built in 1962. The reservoir's average sedimentation rate data for the period 1962 – 2008 (47 years) measured by storage capacity survey are available from a report (CWC, 2015).



### 3. Methodology

This section describes the methods used for computing SE & SY and their associated uncertainties. Following the guidelines given by the Joint Committee for Guides in Metrology (JCGM, 2008), the uncertainties are expressed using standard deviation and reported as percentage of the mean value. To combine uncertainties, the general principle of
adding uncertainties in quadrature is used (Taylor, 1982). The principle assumes that the individual uncertainties are independent.

### 3.1 Estimation of Soil Erosion (SE)

SE is estimated by Revised Universal Soil Loss Equation (RUSLE), which is an empirical model for predicting long-term average rate of SE based on crop system, management techniques, and erosion control practices (Renard et al.,
1991; Kinnell, 2008). The SE is expressed as a function of five input factors (Eq. 1): rainfall and runoff erosivity (R), soil erodibility (K), slope length and steepness (L & S), cover management (C) and support practice (P) (see Annotation for their units). These input factors vary considerably from storm to storm, but their effects on the estimation of SE tend to be averaged over extended periods (Wischmeier and Smith, 1978). The methodology for determining the input factors and their uncertainties at each evaluation cell is described below.

$$SE = R\,K\,L\,S\,C\,P \tag{1}$$

### 3.1.1 Rainfall and Runoff Erosivity Factor (R)

The R factor quantifies the raindrop impact and gives information about the amount and rate of runoff likely to be associated with the rain. The R factor can be obtained by estimating rainfall kinetic energy from rainfall intensity data (Wischmeier and Smith, 1978). As rainfall intensity data are not easily available, empirical equations have been
proposed to calculate R factor from the readily available average annual rainfall data (denoted by r). In this study, we selected equation by Babu et al. (1978) developed using the rainfall data from various meteorological stations in India (Eq. a in Table 2). The uncertainty in the estimate of R factor arises from model error in the Babu et al.'s equation and variability in observed average annual rainfall ($\delta r$). Since model error for Babu et al's equation is not available, the uncertainty in R ($\delta R$) is estimated solely based on observed rainfall variability (Eq. b in Table 2).

### 3.1.2 Soil Erodibility Factor (K)

This factor represents the susceptibility of soil to erosion due to rainfall and runoff. The K factor is usually obtained from one of the many empirical equations (Wischmeier and Smith, 1978; Declercq and Poesen, 1991; Van der Knijff et al., 2000) that relate it to soil properties like organic matter percentage, soil texture, and soil permeability. In this study, the equation proposed by Wischmeier and Smith (1978) and shown as Eq. c in Table 2 is used because all the required
input parameters for this equation are available for the study basin. The uncertainty in the K factor can be due to



uncertainty in the measurement of soil properties and uncertainty in the model that relates soil properties to K factor. Since the measurement uncertainties for soil properties are not available for the study basin, only the model uncertainties as given by Wischmeier and Smith (1978) are considered (Table 2).

### 3.1.3    Slope-Length factor (L) & Slope Steepness factor (S)

5    The L & S factors represent the effect of topography on SE. They are usually presented as a single factor (LS factor) that represents the ratio of soil erosion for the given conditions to the soil erosion from an experimental plot of slope length 22.13 m and slope steepness 9%. This study employs the method proposed by Desmet and Govers (1996) for determining L factor (Eq. d in Table 2). The method calculates L factor by considering the flow accumulation at each cell obtained from DEM. The S factor depends only on the local slope. Many empirical equations are available for 10    estimating the S factor (Wischmeier and Smith, 1978; McCool et al., 1987; Moore and Wilson, 1992; Nearing, 1997). Here in, the equation proposed by McCool et al. (1987) is adopted because of its popularity and versatility (Eq. f in Table 2).

Monte Carlo simulations are usually employed for quantifying uncertainties in LS factor (Biesemans et al, 2000; Catari, 2010; and Tetzlaff et al., 2013). In this method, multiple realizations of DEM are generated based on pre-specified error 15    rate in DEM elevation, and LS factor is calculated for every realization. The variability in LS factors over multiple realizations provides a measure of uncertainty in the LS factor arising due to uncertainties in DEM. The DEM errors are sometimes modeled as an auto-correlated random field (Biesemans et al, 2000); however, in absence of information about the spatial structure of the DEM errors, they are either modeled as independent errors (Tetzlaff et al., 2013) or some simplified assumptions are made on their spatial structure using spatial filters (Catari, 2010). The independent 20    assumption gives the worst-case scenario of DEM uncertainty effects (Wechsler and Kroll, 2006). The Monte Carlo simulations are effective for small size basins, but become tedious for large basins (Tetzlaff and Wendlend, 2012). This study uses the information on geo-location error ($\delta \Delta x$) and elevation error ($\delta \Delta h$) available for the DEM, and applies first-order uncertainty analysis to estimate uncertainties in L ($\delta L$) and S ($\delta S$) factors (Eqs. e and g in Table 2). The method assumes that the DEM errors are uniform in space and are independent.

25    The uncertainty in L factor also depends on uncertainty in specifying the value of variable slope exponent (m). Here in, the uncertainty in m is modeled as a Type B standard uncertainty assuming symmetric triangular distribution (JCGM, 2008, page 11 to 18) over the range of value (0.3 – 0.5). The combined uncertainty in LS factor ($\delta LS$) is obtained by adding $\delta L$ and $\delta S$ in quadrature.

### 3.1.4    Cover and Management (C) & Support Practice Factor (P)

30    The C factor is the ratio of soil loss from a given land use class to the corresponding loss from an experimental plot having "clean-tilled and continuous fallow" land use condition. The P-factor is the ratio of soil loss from a land with





given support practice to the corresponding loss from an experimental plot having an agricultural practice of "upslope and downslope tillage." The C and P factors for a cell are obtained from reference tables (Morgan, 2009; FAO, 1978) that provide a range for given land use and agricultural practices. Reference values of C & P factors for the classes of land use and agricultural practices considered in this study are given in Tables 3 & 4, respectively. Since RUSLE model

is not applicable for glacial erosion and channel processes, the C factor for snow and water covered cells is taken as zero.

For an agriculturally dominated basin, the C factor varies seasonally depending upon the cropping cycle. The seasonality in C factor is incorporated in RUSLE by taking a weighted average of C values during different seasons, where weights are proportional to the R factor (Vanoni, 1975). The study basins has two type of cropping patterns - (1)

double & triple cropping pattern in which crops are grown almost all year round, and (2) single cropping (Rabi/ Kharif/ Zaid) system in which the crops are grown only for a season. Since the farms with single cropping pattern are fallow during the non-growing season they are attributed with wider range of C factor $(0.3 - 1)$ than the farms with double & triple cropping pattern $(0.3 - 0.5)$.

During field visits, we observed that terrace cropping and strip cropping are practiced in most of the upper mountainous

region & lower alluvial plains, respectively. For calculating SE, it is assumed that all the upper mountainous region & lower alluvial plains have same cropping practice, and are assigned same range for the P factor. The uncertainty in C ($\delta$C) and P ($\delta$P) factors are obtained by Type B evaluation of standard uncertainty, assuming symmetric triangular distribution over the range of value in a given class of land use and agricultural practice (JCGM, 2008), presented in Table 2 (Eqs. i & j). The combined effect of land use and land cover is represented by CP factor, which is a product of

C and P factors. The uncertainty is CP factor ($\delta$CP) is calculated by adding $\delta$C and $\delta$P in quadrature.

Finally, assuming that uncertainties in individual factors are independent, they are added in quadrature to calculate relative uncertainty in the estimate of SE for each cell as –

$$\frac{\delta SE}{SE} = \sqrt{\left(\frac{\delta R}{R}\right)^2 + \left(\frac{\delta K}{K}\right)^2 + \left(\frac{\delta LS}{LS}\right)^2 + \left(\frac{\delta CP}{CP}\right)^2} \qquad (2)$$

### 3.2 Estimation of Sediment Delivery Ratio (SDR)

SDR is estimated by the following empirical equation developed by Sharda and Ojasvi (2016) for north Indian River basins.

$$SDR = 1.817 \times A^{-0.132} \qquad (3)$$

The equation is developed based on reservoir sedimentation (CWC, 2015) and soil erosion data (Sharda, 2009; NAAS, 2010) from sixteen large reservoir basins (basin area greater than 1,000 km$^2$) located in north India. The equation was



fitted by the ordinary least squares method in the logarithmic domain. An expression for uncertainty in SDR prediction ($\delta$SDR) that accounts for model error ($\delta$SDR$_{model}$) and uncertainty in the calculation of basin area ($\delta$SDR$_{input\ data}$) derived by using the first-order uncertainty analysis is given by Eq. n in Table 2.

### 3.3 Computation of Sediment Yield (SY)

The SY at a location is estimated by multiplying annual average gross SE and SDR of a drainage basin whose outlet is the estimation point. The uncertainty in SY is computed as the standard deviation of the SY distribution obtained by 1000 Monte Carlo simulations for which gross SE and SDR are simulated using the following distributions -

(i) The gross SE is the sum of average annual SE at all the cells in the basin. Assuming weak form of central limit theorem to be applicable (i.e. probability distribution of annual average SE at all cells in the basin are non-
independent and non-identical), the gross erosion is considered to have a normal distribution with mean and variance equal to sum of means and variances of SE at individual cells, respectively. A truncated version of the normal distribution in the range [0, ∞) is used to avoid negative values of gross SE during Monte Carlo simulations.

(ii) The SDR of the basin is assumed to follow lognormal distribution with mean and standard deviation given by Eqs.
(k) and (n) in Table 2, respectively.

In addition to Monte Carlo simulations, uncertainty in SY is also estimated by first-order uncertainty analysis.

To account for the effect of a dam on SY, we used the method proposed by Sharda and Ojasvi (2016) which assumes that a sufficiently large dam on a river (termed terminal dam) entraps all the sediment carried by the river into its reservoir. Therefore, the gross SE at a location downstream of a terminal dam is estimated from "free basin area" (total
basin area – reservoir catchment area) instead of total basin area. The SY is then calculated as the product of gross SE from the free basin area and SDR of the entire basin.

In this study, annual average SY is estimated at two locations – Nanak Sagar dam for the period 1962-2008 and Husepur gauging station for the period 1987-2002. The Nanak Sagar dam, which lies upstream of the Husepur station, is treated as a terminal dam.

### 3.4 Step by step procedure

The following steps are followed to estimate the values and corresponding uncertainties of SE, SDR and SY. The steps are depicted by a flowchart in Fig. 3.

(i) The Garra River basin boundary is derived from SRTM data based on D8 flow direction algorithm (O'Callaghan and Mark, 1984). Hydrological correction of DEM is done by filling of pits in topography (Tarboton et al., 1991).



Flow accumulations, flow directions, drainage networks and local slopes are calculated from corrected DEM of the Garra River basin.

(ii) The R, K, L, S, C & P factors and their uncertainties are calculated by using the equations given in Table 2 and explained in the previous sections.

(iii) Spatially distributed SE averaged over the study period is estimated by applying RUSLE equation. The uncertainties in individual factors of RUSLE are propagated by first-order uncertainty analysis (Table 2 and Eq. 2) to calculate uncertainty in the estimated SE at each cell. Monte Carlo simulations are used to predict the distribution of SE.

(iv) Mean and variance of gross SE for a basin is estimated by summing up mean and variance of SE at all the cells in the basin.

(v) SDR for a basin is modeled as a lognormal distribution with mean and standard deviation estimated from Eqs. (k) and (n) in Table 2, respectively.

(vi) SY at the Nanak Sagar dam (NSD) and Husepur guaging station (HGS) are estimated by Mote Carlo simulations. 1000 values of gross SE (normal distribution) and SDR (lognormal distribution) are generated and multiplied with each other to simulate 1000 values of SY. The uncertainty is SY is reported as the standard deviation of the simulated SY values.

(vii) The estimated annual average SY at NSD (1962-2008) and HGS (1987-2002) are compared with observed values to assess the suitability of the proposed methodology.

## 4. Results and discussion

### 4.1 Soil erosion (SE)

First, the results of RUSLE factors are presented, which are followed by results of SE estimation. The results presented are the average value during the entire study period (1962 – 2008).

#### 4.1.1 R factor

The R factor (Fig. 4a) follows the same spatial pattern as average annual rainfall (Fig. 2d). It is high in the upper mountainous region where average annual rainfall exceeds 1,000 mm. The factor gradually reduces in the lower alluvial parts and attains the minimum values near the basin outlet. The uncertainty in R factor that stems from variability in annual rainfall varies in a relatively narrow range of 3.4% to 6.7% (Fig. 5a).

#### 4.1.2 K factor

The soil map (Fig. 2c) shows the presence of sand and sandy loam soil close to the main channel and in forested patches of the basin; the rest of the basin is covered primarily with loam. Typically, the loam is more susceptible to erosion than sand & sandy loam, which is reflected in higher values of K factor (Fig. 4b). The upper mountainous region shows



highest values of K factor because the loam in this region has higher silt content (~ 40%) compared to loam in lower alluvial plains (~ 30%). The magnitude of model uncertainty in $\delta$K is constant for the basin, hence, the percentage uncertainty is lower for cells with larger K factor (upper mountainous region) than for cells along the main channel that have low value of K factor. The uncertainties vary from 5.4% to 9.6% (Fig. 5b)

### 4.1.3    LS factor

The LS factor shows considerable variations mainly in the upper mountainous region where its value ranges from 20 to 2,500 (Fig. 4c). The larger values and higher variation in the upper mountainous region can be attributed to the steeper slopes (S factor) and its varying topography. The LS factor is also relatively high for cells close to the stream mainly because of the large contributing area (L factor). For rest of the basin, the LS factor is small (< 20) and shows little variability. The uncertainty in LS factor also shows a considerable variation. The magnitude of uncertainty is significant for cells near the channel and upper mountainous region. However, these cells have least percentage uncertainty (< 2%) because of higher magnitude of LS factor (Fig. 5c). The percentage uncertainties in the rest of the basin vary between 2% to 12% (Fig. 5c).

### 4.1.4    CP factor

The spatial map of CP factor (Fig. 4d) resembles distinct land use land cover (LULC) features present in the basin. The factor is zero for snow and water covered cells. It attains a low value (< 0.1) for forested cells (16% of basin area), intermediate values (0.2 – 0.3) for urban cells (0.4% of basin area), and highest values (> 0 .4) for cropland cells (71% of basin area). The uncertainty in CP factor also varies according to LULC type and crop practice class as shown in Tables 3 & 4. The percentage uncertainty varies 8.2% to 13.6% (Fig. 5d).

### 4.1.5    Soil erosion (SE)

Finally, results of all the factors described in the preceding subsection are combined by using Eqs. (1) and (2) to obtain SE map (Fig. 4e) and its uncertainty (5e), respectively. The two distinct geomorphic settings in the basin – the upper mountainous region and the lower alluvial plains show significant differences in SE. The factors governing the rate of SE in these two settings are compared in Figs. 6 & 7. The SE is highest in the upper mountainous region (SE > 40 t/ha/y; severe category). For the cells near the channels the rate of erosion falls in the zone of very high (20 to 40 t/ha/y) and severe (> 40 t/ha/y) category. Other parts of the basin have moderate (<10 t/ha/y) to high (10 – 20 t/ha/y) SE rate. The average rate of SE for the entire Garra basin is 20.4 t/ha/y (very high), whereas for upper mountainous region and lower alluvial plains the value are 84.4 t/ha/y (severe) and 17.7 t/ha/y (high), respectively.

The upper mountainous region has higher values of R, K and LS factors than the lower alluvial plains. A significant portion of the alluvial plains has cultivated land where the agricultural practices tend to make the soil more susceptible to sheet erosion during rainfall. Hence, the CP factor is higher for the alluvial plains. Nevertheless, the higher erosion



rates in the mountainous region can be attributed mainly to the higher values of LS factor due to steeper slopes, shown in Fig 7.

The uncertainty map of SE rate (Fig 5) reflects the spatial distribution of uncertainty in individual factors. The uncertainty tends to be high for sandy loam and sandy soil patches in the basin. The percentage uncertainty in upper mountainous region is lower (16.5%) than alluvial plains (20.5%). However, uncertainties in the magnitude of erosion rate are higher for mountainous region (13.9 t/ha/y) than alluvial plains (3.6 t/ha/y). The magnitude and percentage uncertainty of RUSLE factors and SE rate averaged over the entire basin is very similar to that of alluvial plains that forms the major portion of the basin (95%; Fig. 6b).

Figure 6a shows the distribution of SE at two representative cells in the upper mountainous region and lower alluvial plains obtained by the Monte Carlo simulations. The cell in the mountainous region has higher value of SE and its distribution has wider spread compared to that of the cell in the alluvial plains. Both the distributions are positively skewed, although the magnitude of coefficient of skewness is small (0.11 for mountainous regions and 0.13 for alluvial plains). Table 5 compares the uncertainties in RUSLE factors and SE reported in the literature, and those obtained in the present study. The reported uncertainties in SE have a wide range that encompasses the uncertainty range estimated in the present study. The backward uncertainly propagation method uses observed data and thus represents true uncertainty. The forward method gives an approximation of the true uncertainty, and usually under predicts the true value.

## 4.2 Sediment Delivery Ratio (SDR) and Sediment Yield (SY)

The SDR and its uncertainty for Nanak Sagar dam (NSD) and Husepur gauging station (HSG) are 0.80 (4.40%) and 0.57 (4.81%), respectively. For both the locations, the SDR model uncertainty component ($\delta SDR_{model}$) dominates total SDR uncertainty ($\delta SDR_{model} > 0.95\ \delta SDR$). The gross SE at the NSD and HSG sties and its uncertainty (reported as coefficient of variation in parenthesis) are $10.1\times10^5$ t/y (16.63%) and $13.8\times10^6$ t/y (20.65%), respectively. The SY and its uncertainty estimated by the first-order uncertainty analysis at NSD and HSG sites are $8.0\times10^5$ t/y (17.32%) and $7.9\times10^6$ t/y (21.21%), respectively. Figure 8 shows the distribution of SY at the two sites obtained by the Monte Carlo simulations. The distributions at both the sides are positively skewed. The standard deviations of the simulated SY at the two sites are almost equal to that obtained from the first-order uncertainty analysis. The 90% confidence interval for the SY based on the Monte Carlo simulations at NSD and HSG are ($5.8\times10^5$ t/y, $10.4\times10^5$ t/y) and ($5.2\times10^6$ t/y, $10.7\times10^6$ t/y), respectively. The observed SY at the two sites are $6.4\times10^5$ t/y and $7.2\times10^6$ t/y, and they lie within the 90% confidence interval of the estimated SY.





## 5. Limitations

This study presents a methodology for quantifying uncertainty in the estimate of SE and SY for ungauged basins based on RUSLE-SDR approach. Uncertainties in SE and SY arise from uncertainties in data, model and due to stochastic nature of the soil erosion process. Like most of the previous studies (referred in Section 1), the proposed methodology

accounts for only those sources of uncertainties that are available or could be quantified easily. For example, models/equations used for estimating R and LS do not provide sufficient details to ascertain model uncertainties, hence only data uncertainties are accounted for. On the other hand, uncertainties in data need for estimating K and CP factors are not available. Hence, only model uncertainties are considered. Similarly, structural uncertainty in the RUSLE-SDR approach could not be quantified due to unavailability of sufficient information. Thus, the proposed methodology does

not account for certain sources of uncertainties leading to under estimation of SE and SY prediction uncertainty. Further, the proposed methodology assumes spatial independence of certain RUSLE factors like R, L, S, C and P factors, resulting in further underestimation of SE and SY prediction uncertainty.

We demonstrated the proposed methodology by applying it to Garra River basin. The basin has data restrictions that are typical of River basins in India. The spatial distributions of SE and SY for the study basin are obtained by using land

use land cover data for 2005, which may not be a true representation of basin conditions during the study period (1962 – 2008). Further, the analysis presented uses gridded daily rainfall data available at a spatial resolution of 0.25° x 0.25° obtained by interpolating rain gauge observations. The coarser spatial resolution of the data is not sufficient to capture spatial variability of rainfall in the basin. In addition, the gridded rainfall data may have large interpolation errors, which are not accounted, because as they are not available for the study basin. The uncertainty in topographic features

is estimated by using SRTM DEM errors reported at a continental scale (Asia region). This continental scale error may be very different from the local scale errors in the basin. In addition, there are limitations inherent in the RUSLE based approach (Vanoni, 1975). It accounts for sheet and rill erosion but ignores gully, bank and channel erosion processes. Further, it predicts long-term average erosion but cannot make event-based or seasonal predictions.

In spite of many limitations, the proposed framework for quantifying and propagating uncertainties in SE and SY

appears promising, particularly for ungauged basins.

## 6. Concluding Remarks

The main objective of this work is to present a methodology for quantifying uncertainties in the estimates of soil erosion (SE) and sediment yield (SY). A systematic procedure is provided for evaluating and propagating uncertainties in a RUSLE-SDR based approach for SE and SY prediction. Expressions for uncertainty propagation are derived using

first-order uncertainty analysis making the proposed methodology viable even for large river basins. The novelty of the work lies in presenting a unified framework for quantifying uncertainties in SE and SY that is applicable to ungauged





basins with storage structures. The methodology is demonstrated by applying it on the Garra River basin. A summary of the work and conclusion derived from it are listed below.

(i)    Various datasets needed for applying RUSLE model are collected and used for estimating spatially distributed annual average SE rates in the study basin.

(ii)    SE in the basin is very high ($20.4 \pm 4.1$ t/ha/y) with higher values in the upper mountainous region ($84.4 \pm 13.9$ t/ha/y) than lower alluvial plains ($17.7 \pm 3.6$ t/ha/y).

(iii)    The LS and CP factors govern the magnitude of soil erosion and its uncertainty in upper mountainous region and lower alluvial plains, respectively.

(iv)    The sediment delivery ratio (SDR) values for Nanak Sagar dam (NSD) and Husepur gauging station (HSG) are
estimated to be 0.80 and 0.57, respectively, with about 5% uncertainty in both the estimates.

(v)    SY at NSD and HSG are estimated to be $8.0 \pm 1.4 \times 10^5$ t/y (CV = 17.32%) and $7.9 \pm 1.7 \times 10^6$ t/y (CV = 21.21%) respectively. The observed values at the two sites are $6.4 \times 10^5$ t/y and $7.2 \times 10^6$ t/y respectively, and they lie within estimated 90% confidence interval. The results suggest that the proposed approach could be effective for sheet or rill erosion dominated Himalayan River basins like the Garra basin.

(vi)    The uncertainty in SY derived from Monte Carlo simulations and first-order uncertainty analysis are very similar. The distributions of SY at both the sites are positively skewed, although the magnitude of coefficient of skewness is small.

(vii)    Not all sources of uncertainties could be accounted in the study because of limited available information. Hence, the estimated uncertainties in SE and SY are underestimation of true uncertainties. Review of
uncertainties reported in the literature suggests that true uncertainty can be much higher than the predicted uncertainty. However, in absence of long records of observed SY, the quantification of true uncertainty remains a challenge.

**Acknowledgements**

The authors acknowledge the financial support provided by the Ministry of Earth Sciences, New Delhi under the Indo-UK joint programme on Changing Water Cycle. The first author also acknowledges the Ph.D. studentship from IIT Kanpur.

**Annotations**

C, the cover and management factor is the ratio of soil loss from an area with specified cover and management to that of
an identical area in tilled continuous fallow

ΔC, difference between upper and lower limit of C factor

CV, the coefficient of variation





CS, the coefficient of skewness

$\Delta h$, maximum difference in the elevation between the given cell and its neighbor cells (8 neighboring cells in D8 algorithm)

$\delta\Delta h$, elevation error in DEM (3.17 m)

K, the soil erodibility factor, expressed in the units of ton ha hr $MJ^{-1}mm^{-1}$ $ha^{-1}$

L, the slope length factor, is the ratio of soil loss from the field slope length to that from a 22.1meters length under identical conditions

$\lambda$, field slope length in meters

$\delta\lambda$, uncertainty in field slope length

M, Particle-size parameter [% silt x (100 - % clay)]

m, variable slope-length exponent (0.3 – 0.5)

$\delta m$, uncertainty in variable slope-length exponent

n, number of cells contributing to one cell

OM, organic matter content (%)

P, the support practice factor, is the ratio of soil loss with a support practice like contouring, strip cropping, or terracing to that with straight-row farming up and down the slope

p, permeability class (rapid = 1, moderate to rapid = 2, moderate = 3, slow to moderate = 4, slow = 5, very slow = 6)

$\Delta P$, difference between upper and lower limit of P factor

r, the average annual rainfall in mm

R, the rainfall erosivity factor, expressed in the units of MJ mm $ha^{-1}$ $hr^{-1}$ $y^{-1}$

S, the slope steepness factor, is the ratio of soil loss from the field slope gradient to that from a 9-percent slope under otherwise identical conditionssc, soil structure code (very fine granular = 1, fine granular = 2, coarse granular = 3, blocky, platy or massive = 4)

SE, the computed soil erosion per unit area expressed in t $ha^{-1}$ $y^{-1}$

SY, the sediment yield at a location in the basin, expressed in t $y^{-1}$





θ, slope of the terrain in degrees

Δx, distance between the given cell and the neighboring cell having maximum elevation difference

δΔx, geo-location error in DEM (5.17 m)





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





**Tables**

**Table 1 Data used in the study and their specifications.**

| Sr. No. | Input Data | Name & Agency | Specifications | Reference |
|---|---|---|---|---|
| 1 | Digital Elevation Model (DEM) | SRTM (Shuttle Radar Topography Mission) | 90 m spatial resolution; year 2001; database version 4.1 | Jarvis et al. (2008) |
| 2 | Land use and land cover | National Remote Sensing Center (NRSC) | 1:50,000 scale; year 2005 | NRSC (2006) |
| 3 | Soil | | 56 m spatial resolution; year 2005 | National Bureau of Soil Survey (NBSS) |
| 4 | Annual average rainfall | India Meteorology Department (IMD) | 0.25° spatial resolution; daily gridded rainfall data set from 1901 to 2013) | Rajeevan and Bhate (2009) |

5 **Table 2 Equations and their references for estimating RUSLE factors, sediment erosion, and sediment delivery ratio (SDR). Expressions are also given for quantifying and propagating uncertainty based on first-order analysis.**

| Factor | Estimation of Factor | Estimation of Uncertainty |
|---|---|---|
| R | $R = 79 + 0.363 \times r$ **(a)** <br> (Babu et al. 1978) | $\delta R = 0.363 \times \delta r$ **(b)** |
| K | $100K = 2.1 \times 10^{-4} \times (12 - OM) \times M^{1.14} + 3.25 \times (sc - 2) + 2.5 \times (p - 3)$ **(c)** <br> (Wischmeier and Smith, 1978) | $\delta K = \text{Calculated} - \text{Measured}$ <br> $= 0.0026$ (65 % Confidence Interval) <br> (Wischmeier and Smith, 1978) |
| L | $L = \dfrac{(\lambda_{i-1} + D)^{m+1} - (\lambda_{i-1})^{m+1}}{D(22.13)^m}$ **(d)** <br> (Wischmeier and Smith, 1978; Desmet and Govers, 1996) | $\dfrac{\delta L}{L} = \sqrt{\left(\dfrac{m}{\Delta x}\delta\Delta x\right)^2 + (\ln(m+1)\,\delta m)^2}$ **(e)** <br> where, $\delta m = \dfrac{\Delta m}{2\sqrt{6}}$ (Triangular distribution) |
| S | $S = 10.8 \times \sin\theta + 0.03$ for slope < 9%; <br> $S = 16.8 \times \sin\theta - 0.05$ for slope $\geq$ 9% **(f)** <br> (McCool et al., 1987) | $\delta S = 10.8 \times \cos\theta \times \delta\theta$ slope < 9% , <br> $\delta S = 16.8 \times \cos\theta \times \delta\theta$ slope $\geq$ 9% **(g)** <br> $\delta\theta = \sqrt{\left(\dfrac{\delta\Delta h}{\Delta x \times \left(1 + \left(\dfrac{\Delta h}{\Delta x}\right)^2\right)}\right)^2 + \left(\dfrac{-\delta\Delta x}{\Delta h \times \left(1 + \left(\dfrac{\Delta x}{\Delta h}\right)^2\right)}\right)^2}$ **(h)** |
| C | Reference tables (Morgan, 2009; | $\delta C = \dfrac{\Delta C}{2\sqrt{6}}$ (Triangular distribution) **(i)** |





| P | FAO, 1978) | $\delta P = \frac{\Delta P}{2\sqrt{6}}$ (Triangular distribution) **(j)** <br><br> (JCGM, 2008) |
|---|---|---|
| SDR | $SDR = 1.817 \times A^{-0.132}$ **(k)** <br><br> (Sharda and Ojasvi, 2016) | $\delta SDR_{model} = \sqrt{(\exp(se^2) - 1) \times \exp(2\ln(SDR) + se^2)}$ **(l)** <br> where, $se$ = standard error (0.048) <br> $\delta SDR_{input\ data} = 0.24 \times A^{-1.132} \times \delta A$ **(m)** <br> where $\delta A = n \times 2\Delta x \times \delta \Delta x$ <br> $\delta SDR = \sqrt{(\delta SDR_{model})^2 + (\delta SDR_{input\ data})^2}$ **(n)** |

**Table 3 C factor for different land use and land cover classes along with their uncertainties.**

| LULC Class | C Factor Range | Mean Value | Uncertainty |
|---|---|---|---|
| Forest | 0.001 – 0.002 | 0.0015 | 13.61% |
| Grassland | 0.01 – 0.02 | 0.015 | 13.61% |
| Urban | 0.05 – 0.1 | 0.075 | 13.61% |
| Plantation/Orchard | 0.1 – 0.3 | 0.2 | 20.41% |
| Crops (Double & Triple) | 0.3 – 0.5 | 0.4 | 10.21% |
| Crops (Kharif, Rabi & Zaid) | 0.3 - 1 | 0.65 | 22% |
| Wasteland | 0.4 – 0.6 | 0.5 | 8.16% |
| Water/Snow | 0 | 0 | 0 |

**Table 4 Different cropping practice (P) factor for various cropping practice along with their uncertainties.**

| Crop Practice | P-Factor Range | Mean value | Uncertainty |
|---|---|---|---|
| Strip Cropping | 0.6 – 0.9 | 0.75 | 8.16% |
| Terrace Cropping | 0.35 – 0.45 | 0.4 | 5.1% |
| Other Areas | 1.0 | 1.0 | 0% |





**Table 5 Comparison of uncertainties in RUSLE factors and soil erosion (SE) reported in the literature and those obtained in the present study. The present study employs forward uncertainty propagation for the Garra river basin.**

| Factor | Reference | Range of uncertainty | Scale | Method | Present study |
|--------|-----------|----------------------|-------|--------|---------------|
| R Factor | Catari 2010 | 7 – 16 % | Basin | Forward | 3.4 – 6.7 % |
| | Catari et al. 2011 | 8.9 – 10 % | Basin | Forward | |
| | Wang et al. 2002b | 30 – 40 % | Basin | Forward | |
| K Factor | Catari 2010 | 5 – 90 % | Basin | Forward | 5.4 – 9.6 % |
| | Parysow et al. 2003 | 25 – 35 % | Plot | Forward | |
| | Wang et al. 2001 | 5 – 25 % | Basin | Backward | |
| LS Factor | Mondal et al. 2016 | 3 – 12 % | Basin | Forward | 2 – 12 % |
| | Wang et al. 2002a | 0 – 15 % | Plot | Forward | |
| CP Factor | Hession et al. 1996 | 10 % | Plot | Backward | 8.2 – 13.6 % |
| | Tetzlaf & Wendland 2012 | 23 % | Basin | Backward | |
| | Tetzlaff et al. 2013 | 23 % | Basin | Backward | |
| Soil Erosion | Biesemans et al. 2000 | 1.7% | Basin | Forward | 11 – 29 % |
| | Catari 2010 | 10 – 20 % | Basin | Forward | |
| | Hession et al. 1996 | 40 – 50 % | Plot | Forward | |
| | Risse et al. 1993 | 57 – 62 % | Plot | Backward | |
| | Tetzlaff et al. 2013 | 34% | Basin | Forward | |
| | Tetzlaf & Wendland 2012 | 34% | Basin | Forward | |



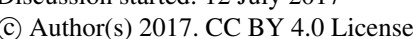



**Figure 1LANDSAT image (1999-2000 ) in False color composite showing the Garra River basin. The major neighboring rivers (Ganga & Ramganga), location of major cities, the gauging station (Husepur) and the major water structure (Nanak Sagar Dam) are also shown.**





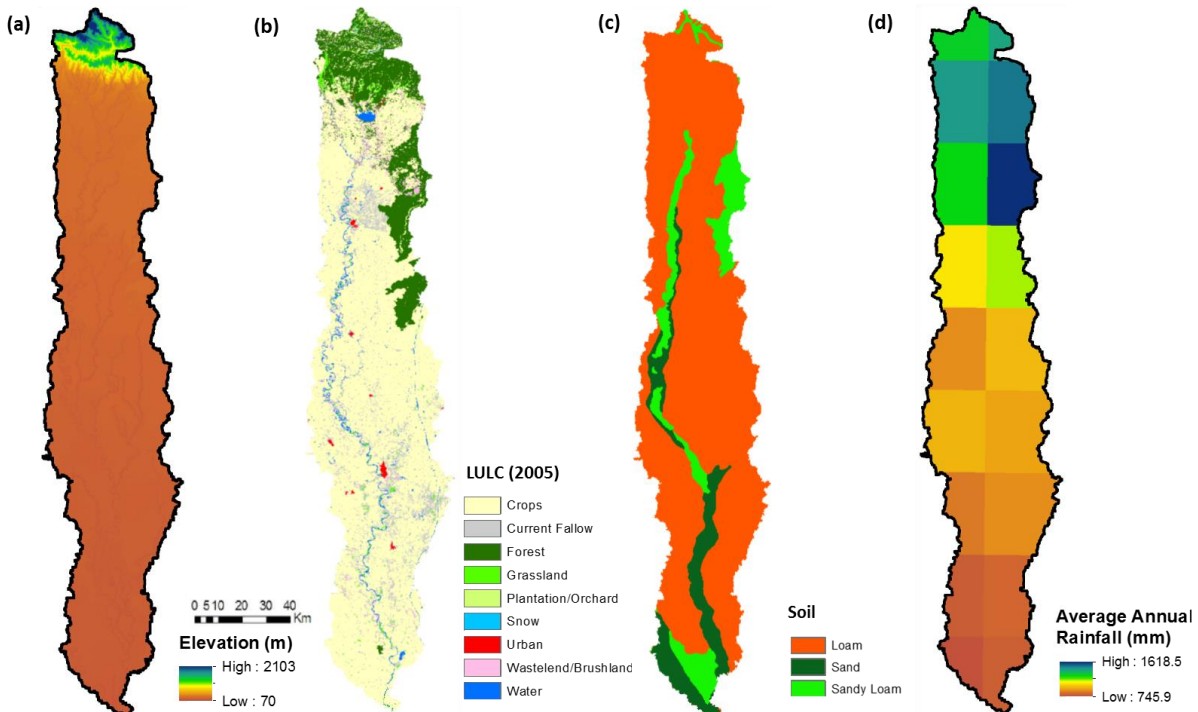

**Figure 2 (a) Elevation (SRTM, 90 m),(b) LULC and (c) Soil data from NRSC (2005) and (d) Average annual rainfall (1962 – 2008).**





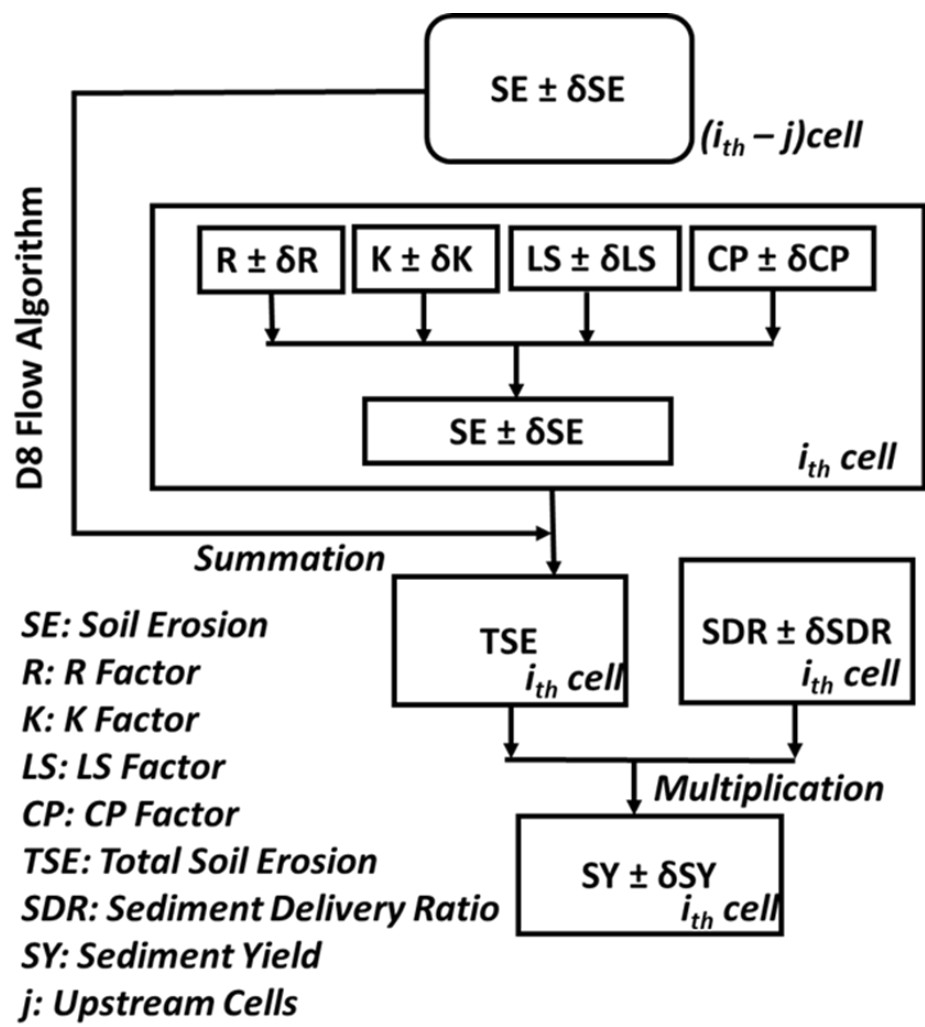

**Figure 3 Approach to estimate soil erosion and sediment yield with associated uncertainties.**




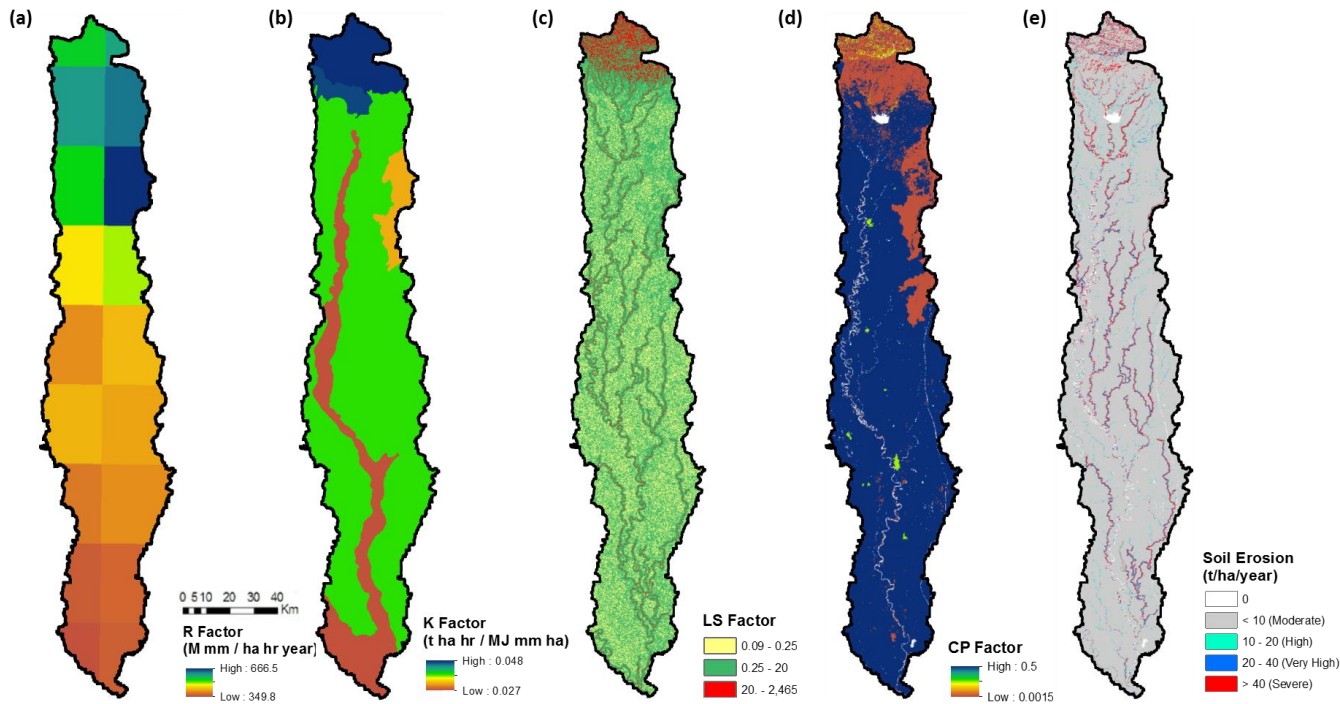

**Figure 4 (a) Rainfall erosivity (R) (b) Soil erodibility (K) (c) Topographic steepness (LS) (d) Crop practice (CP) factors, and (e) Soil Erosion estimation for the Garra River basin.**





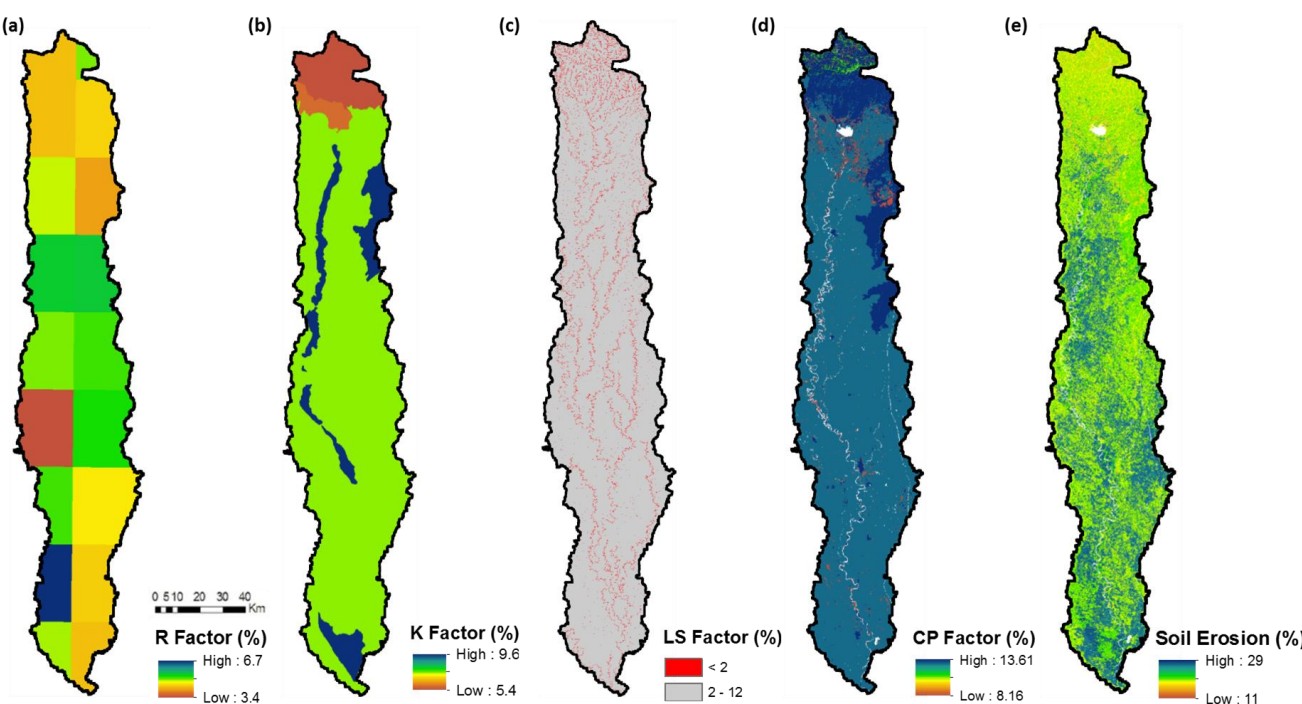

**Figure 5 Percentage uncertainty in (a) Rainfall erosivity  (b) Soil erodibility  (c) Topographic steepness (d) Crop and practice and (e) Soil erosion uncertainty in percentage for the Garra River basin.**





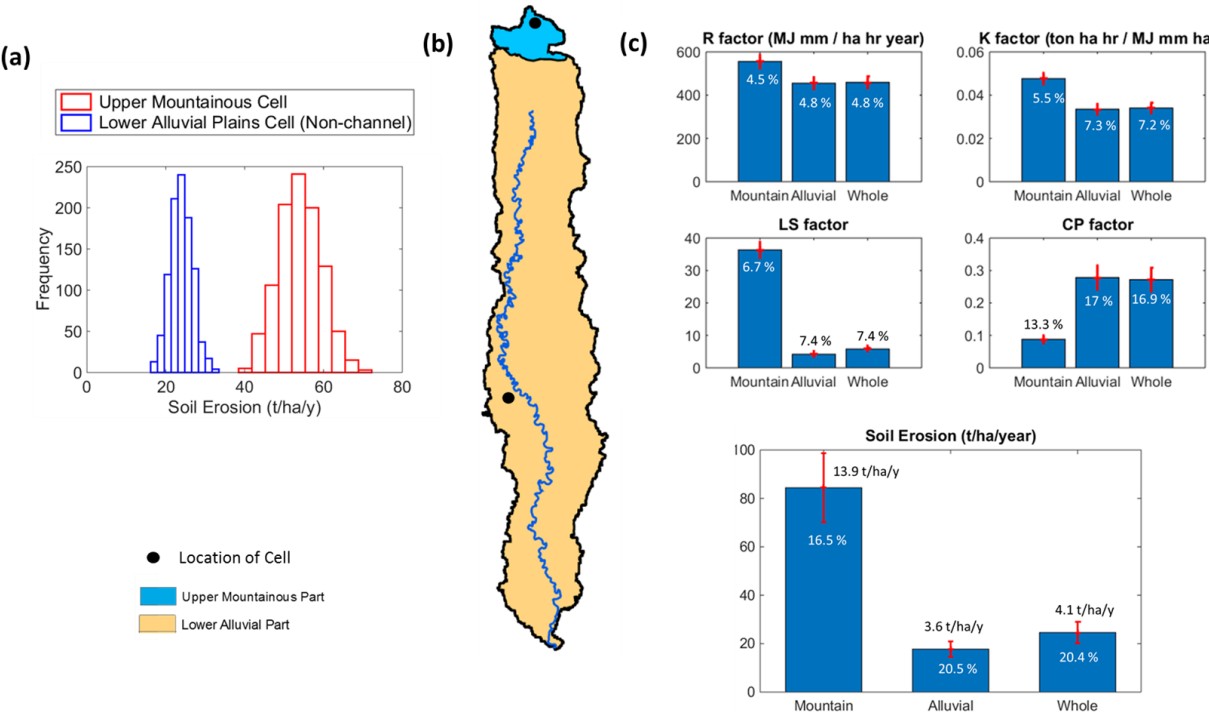

**Figure 6 (a)** distribution of SE at their representative cells in the basin namely, upper mountainous part and lower and lower alluvial part **(b)** Upper mountainous and alluvial plains part of the basin **(c)** comparison between the different factors of RUSLE and SE for both region.




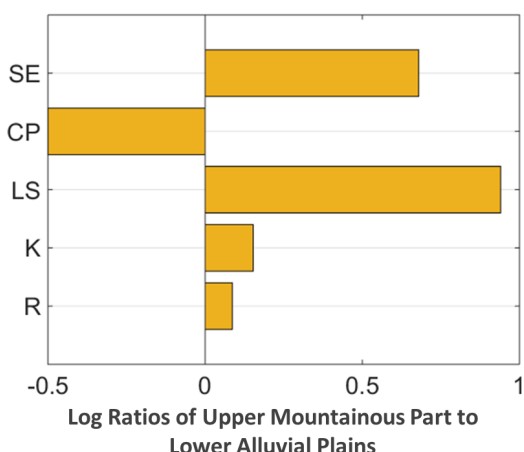

**Figure 7 Comparison of RUSLE factors (R, K, LS & CP) and SE rates (SE) for upper mountainous and lower alluvial plains in the study basin.**

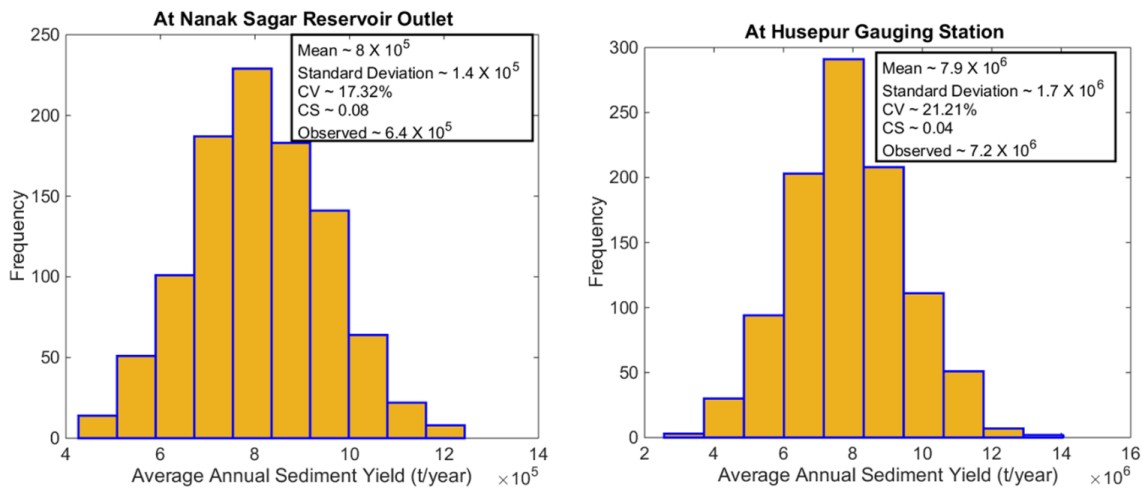

5      **Figure 8 Histogram of annual average sediment yield (SY) at Nanak Sagar dam and Husepur gauging station based on Monte Carlo simulations. The mean, standard deviation, and coefficient of variation of the simulated SY and observed SY at the two sites are given in the legend.**