# Peer review of "Assessment of uncertainties in soil erosion and sediment yield estimates at ungauged basins: an application to the Garra River basin, India"

_Hydrology and Earth System Sciences, 2017_

## Short Comment (SC1) · 27 Jul 2017

HESS – Uncertainties of soil erosion in Garra River, India.

This can be an interesting article if authors carefully address some improvements in uncertainties of erosion factors.

In the current study the erosivity factor (R-factor) has high uncertainties. I am not in favour of functions which estimate erosivity based on annual of monthly rainfall values (you can see the low quality results with large pixels in R-factor). Currently there is an

increasing availability of high temporal resolution rainfall data which allow to estimate rainfall erosivity according to the principles of USLE/RUSL/e. The recent publication and data availability of Global Rainfall Erosivity has demonstrated this and there are about 250 stations with measured R-factor in India.

Regarding soil erodiblity, the recent developments show that also soil structure and Stoniness should be taken into account. Moreover, an additional source of uncertainty has to do with interpolating methods (how did you produce surface maps from the K-factor measurements) and the high organic carbon soils (there is literature about how to interpolate K-factor and how to face the issue of high soil organic carbon).

In the topographic factor, authors do not discuss the pixel size issue. There much higher uncertainty when LS-factor is calculated with pixels of 90m resolution compared too much higher resolution of 25m (all this has been discussed in European application of LS-factor). Moreover, I see values of LS-factor = 2465 .This is impossible for soil coverages.

The cover management factor is the most uncertain in USLE applications. In the manuscript it is not clear (Table 3c) how you got those C-factor ranges and how you calibrate at pixel level? The use of remote sensing on vegetation density may help you on this.

Also how did you find the P-factor values? The literature has quite different values. The first concluding remark is not valid. this is obvious! The soil erosion map could have at least 6-7 classes to show a clear distinction between low erosion , low medium , medium, high , severe, etc (with colours from Green to Red).

Tables should be self-explained. I don' agree with the current structure presenting the equations in the table and having the factors and annotations in separate page. It is not easy for readers.

I call the authors to take into account the above mentioned comments and improve

both the model estimates and the manuscript.

---

## Short Comment (SC2) · 29 Aug 2017

Dear Dr Panagos,

We sincerely thank you for your valuable comments on this manuscript. We have tried our best to incorporate all your suggestions. Detailed answers to the specific questions are given in the following paragraphs.

Comment 1: In the current study, the erosivity factor (R-factor) has high uncertainties. I am not in favor of functions which estimate erosivity based on annual of monthly rainfall values (you can see the low quality results with large pixels in R-factor). Currently there is an increasing availability of high temporal resolution rainfall data which allow to estimate rainfall erosivity according to the principles of USLE/RUSL/e. The recent publication and data availability of Global Rainfall Erosivity has demonstrated this and there are about 250 stations with measured R-factor in India.

**Response 1:** We agree that R factor estimated using high temporal resolution (sub – hourly) dataset based on the principles of USLE/RUSLE is better than that estimated using coarser resolution (monthly or annual) rainfall values. High-resolution rainfall datasets are available for the recent period (example R factor estimated in Global Rainfall Erosivity dataset for India uses hourly data of 250 rain gauges for 2007 - 2015). However, such high-resolution rainfall datasets are not available for the entire study period that starts from 1962 (selected as per the availability of sediment yield records). Since significant annual and decadal variability in the rainfall pattern exists over India, R factor estimated for the recent period may not be a true representative of the study period ranging from 1962 - 2008. Hence, we used IMD dataset for the study that are available at a spatial resolution of 0.25 degree for 1901 - 2013 based on more than 6,000 rain gauge stations over India. Further, the framework proposed for quantifying and propagating uncertainties in SE, SDR and SY estimates, which is the main focus of this paper, is applicable for R factor derived from different resolution datasets based on different principles.

Comment 2: Regarding soil erodibility, the recent developments show that also soil structure and Stoniness should be taken into account. Moreover, an additional source of uncertainty has to do with interpolating methods (how did you produce surface maps from the Kfactor measurements) and the high organic carbon soils (there is literature about how to interpolate K-factor and how to face the issue of high soil organic carbon).

**Response 2:** The present study accounts for soil structure in K factor estimates by using soil structure classification (sc) given by Wischmeier and Smith (1978). The equation (c) in Table 2 provides the equation for K factor estimate, and the structure classification codes are given in the annotation (page 15, line 23).

The Garra basin has primarily three kinds of soil textures: loam, sand and sandy loam (Figure 2c) with negligible amount of gravels. Hence, stoniness is not accounted in the K factor estimates. We will explicitly mention this fact in the revised manuscript.

Soil in the Garra basin has organic matter (OM) less than 0.2%. This value is much smaller than 4%, the maximum range of OM for which Wischmeier and Smith (1978) equation for K factor estimate is applicable.

The K factor is obtained based on NRSC (National Remote Sensing Center) soil dataset available at a spatial resolution of 56m. Thus, no interpolation is performed in estimating K factor map. Instead, the interpolation was performed by NRSC in preparing the soil data. Since NRSC did not provide the interpolation uncertainty, it was not included in the study. This limitation arising due to non-availability of interpolation error in soil data (also present in rainfall and LULC data) will be mentioned in section 5 (Limitations) of the revised manuscript.

Comment 3: In the topographic factor, authors do not discuss the pixel size issue. There much higher uncertainty when LS-factor is calculated with pixels of 90m resolution compared too much higher resolution of 25m (all this has been discussed in European application of LS-factor). Moreover, I see values of LS-factor = 2465. This is impossible for soil coverages.

**Response 3:** In the literature review section, we have given references that discuss the issue of LS factor uncertainty due to cell size variation (page no. 2, line 25). However, this issue was not discussed in length because the objective of this paper is not to study the effect of cell size variation on LS factor uncertainty, but rather to provide a methodology to estimate LS factor uncertainty arising due to errors in DEM (geo-location and elevation errors). Different resolution DEMs obtained from different measurement techniques (remote sensing or ground based survey) may have different geo-location and elevation errors resulting in different LS factor uncertainties. The proposed methodology can be used to estimate LS factor uncertainty irrespective of the DEM resolution or measurement techniques used for its preparation.

Yes, LS factor can not be so high. We made a mistake in plotting the LS values. Figure 1 (a) below shows the correct values of LS factor. The maximum value of LS factor in the study region is 53. Since it was a plotting error, it has no effect on the subsequent results.

Figure 1 (a) Modified LS factor (b) re-classified soil erosion map

Comment 4: The cover management factor is the most uncertain in USLE applications. In the manuscript it is not clear (Table 3c) how you got those C-factor ranges and how you calibrate at pixel level? The use of remote sensing on vegetation density may help you on this.

**Response 4:** Yes, we agree that the cover management (CP) factor is the most uncertain among other RUSLE factors. This is evident in our results shown in Fig 6(c), where the magnitude of CP factor uncertainty is much higher than other factors.

The C factor ranges given in Table 3 are obtained from Morgan (2009; table 6.2, page no. 122). The ranges for different LULC classes were used to estimate uncertainty in C factor by assuming a triangular distribution that spans the entire range of C factor (equation (i) in table 2).

The vegetation density obtained using remote sensing (vegetation indices) can provide an alternative method to quantify C factor and its uncertainty. This will be mentioned in the revised manuscript.

**Comment 5: Also how did you find the P-factor values? The literature has quite different values.**

**Response 5:** The P factor is obtained from Morgan (2009; table 6.3, page no. 123). The table provides P factors for contour and strip cropping based on slope conditions.

Comment 6: (a) The first concluding remark is not valid. this is obvious! (b) The soil erosion map could have at least 6-7 classes to show a clear distinction between low erosion, low medium, medium, high, severe, etc (with colours from Green to Red). (c) Tables should be self-explained. I don't agree with the current structure presenting the equations in the table and having the factors and annotations in separate page. It is not easy for readers.

**Response 6:**

- a) The point is not a concluding remark, but it summarises the work done. We will remove this point from the revised manuscript.
- b) Figure 1(b) in this response shows the soil erosion map with six classes (increased from four in the original manuscript). We hope that the new figure distinguishes different soil erosion areas.
- c) Thanks for the suggestion. In the revised manuscript, symbols will be explained along with the equations in which they are used.

**References:**

Morgan, R. P. C. Soil erosion and conservation. John Wiley & Sons, 2009.

Wischmeier, W. H., & Smith, D. D. Predicting rainfall erosion losses. Agricultural Handbook no. 537, US Department of Agriculture. Science and Education Administration, 1978.

---

## Referee Comment (RC1) · Anonymous Referee #1 · 25 Sep 2017

This manuscript present a methodology in quantifying uncertainties in soil erosion, SDE and SY estimate. The RUSLE-SDR model was employed in the Grra River basin with upper hilly and lower alluvial plains. The manuscript gives general SE and SY for the study area. The topic is interesting, while there are a variety of questions related modeling when there isn't enough calibration and validation.

(1) Page 2, The main objective of the manuscript is presented in lines 10-12. However, there are repeated detailed objectives below. I would suggest a more clear structure for the Introduction section.

[Figure]

(2) As for R in RUSLE, is it a rainfall and runoff erosivity factor according to the original model concept?

(3) In page 5, line 26, the discharge and sediment load records for 16 years are available at the stations. The manuscript should give more details on model calibration and validation.

(4) There is a large reservoir built in 1962, which may play an imporant role in sediment trapping. The sedimentation rate data can be used for sediment yield calibration. The SDR should also consider the effect of reservoir trapping, though the model is empirical.

(5) when compared the annual rainfall and rainfall erosivity, I found the R factor is much lower than the regions with similar rainfall amount, I doubt the proposed the method for R estimation. As well, the very coarse rainfall data might be the dominant factor influencing the simulation results, rather then the R factor itself.

(6) soil map is rough too, I would suggest to do a field survey for sampling, or obtain a relative detailed soil data.

(7) as for LS factor, the maximum value is around 2500, this is extremely high due to the high gradients. This means the LS factor may be overestimated for the steep area, since the RUSLE model was originally developed for estimating soil erosion in relative gentle arable land.

(7) When I saw the data listed in Table 2, the resolution for different data may cause high uncertainties for modeling results. The resolution of the spatial data highly infuence the data quality, such as LS factor, K-factor, C and P factor.

---

## Short Comment (SC3) · 18 Oct 2017

We sincerely thank you for your valuable comments on this manuscript. We have tried our best to incorporate all your suggestions. Detailed answers to the specific questions are given in the following paragraphs.

**Comment 1: Page 2, the main objective of the manuscript is presented in lines 10-12. However, there are repeated detailed objectives below. I would suggest a more clear structure for the Introduction section.**
**Response 1:** As suggested by the reviewer, we have removed lines 10 – 12 in page 2 to avoid any repeatition.

**Comment 2: As for R in RUSLE, is it a rainfall and runoff erosivity factor according to the original model concept?**
**Response 2:** Yes, R factor is used as rainfall and runoff erosivity factor according to the original model concept proposed by Wischmeier and Smith (1978). We have quantified R factor using the relationship between average annual rainfall amount (p) and rainfall & runoff erosivity factor (R) proposed by Babu et al. (1978) based on observed data over India.

**Comment 3: In page 5, line 26, the discharge and sediment load records for 16 years are available at the stations. The manuscript should give more details on model calibration and validation**.
**Response 3:** Monthly discharge and sediment load for 16 years and average sedimentation rate for 40 years are available at Husepur gauging station (HGS) & Nanak Sagar dam (NSD), respectively. We have not performed any calibration for the model but used the parameter values available in the literature. The model was validated by the available sediment yield records at both the stations (HSG & NSD), and the results for the same are given in Figure 8, page 31.

**Comment 4: There is a large reservoir built in 1962, which may play an imporant role in sediment trapping. The sedimentation rate data can be used for sediment yield calibration. The SDR should also consider the effect of reservoir trapping, though the model is empirical.**
**Response 4:**

**(a)** Yes, our results also suggest that the NSD reservoir traps a significant portions of eroded sediment ($6.4 \times 10^5$ tones/year; ~10% of the sediment yield at the basin outlet).
**(b)** We have used the equation given by Sharda & Ojasvi (2016) for North Indian rivers to account for sediment trapping at the reservoir in the sediment yield estimation.
**(c)** As rightly suggested by the reviewer, sediment yield estimate should consider the reservoir trapping. In this work, gross soil erosion for the Garra basin is estimated by extracting the area covered by the reservoir. It is called as gross soil erosion for free basin area (total basin area – reservoir basin area; Sharda and Ojasvi, 2016), which was used along with SDR to estimate sediment yield at the basin outlet. The SDR estimation, however, uses total basin area because the reservoir only traps the sediment but water always flows through it. In other words, at the reservoir outlet, this system is hydrologically connected but sediment connectivity is poor.

**Comment 5: When compared the annual rainfall and rainfall erosivity, I found the R factor is much lower than the regions with similar rainfall amount, I doubt the proposed the method for R estimation. As well, the very coarse rainfall data might be the dominant factor influencing the simulation results, rather than the R factor itself.**
**Response 5:** In this study, we selected equation proposed by Babu et al. (1978) that was developed using observed rainfall data at various meteorological stations in India (Eq. a in Table

2). This equation is based on the linear regression between annual average rainfall amounts and R factor. It is possible that this equation under-estimates the R factor due to its simplification. Also, due to unavailability of rainfall intensity data during the study period, we could not apply intensity based R factor calculation. However, Babu et al. (1978) equation has been widely used for soil erosion predictions in the Indian region (Jain et al, 2010; Kumar et al, 2014; Dutta et al, 2015). Since the aim of this study is to assess uncertainty using easily available datasets and most commonly used equations, we have selected Babu et al's approach for computing R factor and assessing corresponding uncertainties.

**Comment 6: Soil map is rough too, I would suggest to do a field survey for sampling, or obtain a relative detailed soil data.**
**Response 6:** We have used National Remote Sensing Center (NRSC) soil data (1:50,000; 25 m), described in Table 1, page 23. This is the finest resolution soil dataset available for this region and is based on extensive field validation. Given the size of the basin, independent field survey would be a bit complicated to replicate this available dataset. We have re-classified the soil map into soil textural classes namely loam, sand and sandy loam (Figure 2 (c), page 26) for visualization, which is why it looks "rough". The actual dataset has 11 soil classes (shown in Figure 1(b) below) and K factor is estimated for each class.

**Comment 7: As for LS factor, the maximum value is around 2500, this is extremely high due to the high gradients. This means the LS factor may be overestimated for the steep area, since the RUSLE model was originally developed for estimating soil erosion in relative gentle arable land.**
**Response 7:** Yes, the LS factor can not be so high. We made a mistake in plotting the LS values. Figure 1 (a) below shows the correct values of LS factor. The maximum value of LS factor in the study region is 53. Since it was a plotting error, it has no effect on the subsequent results. We have updated this figure in the revised version of the manuscript.

**Comment 8: When I saw the data listed in Table 2, the resolution for different data may cause high uncertainties for modeling results. The resolution of the spatial data highly influence the data quality, such as LS factor, K-factor, C and P factor.**
**Response 8:** The limitation of the datasets arising due to their coarser resolution are explained in the "Limitation" section, page 13. Yes, the coarse resolution datasets may induce uncertainty in modelling results.

[Figure]

Figure 1 (a) Modified LS factor (b) Soil class map

**References:**

Babu, R., Tejwani, K. G., Agarwal, M. C., & Bhushan, L. S. Distribution of erosion index and iso-erodent map of India. Indian Journal of Soil Conservation, 1978.

Dutta, D., Das, S., Kundu, A., & Taj, A. Soil erosion risk assessment in Sanjal watershed, Jharkhand (India) using geo-informatics, RUSLE model and TRMM data. Modeling Earth Systems and Environment, 1(4), 37, 2015.

Jain, M. K., Mishra, S. K., & Shah, R. B. Estimation of sediment yield and areas vulnerable to soil erosion and deposition in a Himalayan watershed using GIS. Current Science, 213-221, 2010.

Kumar, A., Devi, M., & Deshmukh, B. Integrated remote sensing and geographic information system based RUSLE modelling for estimation of soil loss in western Himalaya, India. Water resources management, 28(10), 3307-3317, 2014.

Sharda, V. N., & Ojasvi, P. R. A revised soil erosion budget for India: role of reservoir sedimentation and land-use protection measures. Earth Surface Processes and Landforms, 41(14), 2007-2023, 2016.

Wischmeier, W. H., & Smith, D. D. Predicting rainfall erosion losses. Agricultural Handbook no. 537, US Department of Agriculture. Science and Education Administration, 1978.

---

## Referee Comment (RC2) · Anonymous Referee #2 · 14 Feb 2018

This manuscript is very well written, although there are some inconsistency and issues which I have highlighted in the review report.

I hope other reviewers are able to cover the methodological part of the uncertainty in soil erosion as this is not my field of expertise.

Thank you for this opportunity!

Please also note the supplement to this comment:

[Figure]

https://www.hydrol-earth-syst-sci-discuss.net/hess-2017-383/hess-2017-383-RC2-supplement.pdf

[Figure]

**Supplement:**

The paper "Assessment of uncertainties in soil erosion and sediment yield estimates at ungauged basins: an application to the Garra River basin, India" by Swarnkar et al. highlights the different uncertainties in erosion and sediment yield estimations. Overall, the paper is written very well highlighting the different level of uncertainty and erosion and sediment yield assessment. The results also show the viability of this method for SE and SY estimation in the Garra River basin in India. The paper is very well written and can be published in HESS. However, I have following comments which can be incorporated before the publications:

**Major comments**

- Page 6: Line 20-24: How the spatial variability of rainfall is taken into account. Since the measured rainfall data is from stations, which interpolation method was adopted to spatially distribution the rainfall data for each grid.
- Section 3.4: Step by Step procedure: is not a methodological step. Probably the sentence can just continue without the section.
- 4.1.1: Figure 4(a) suggest the highest rainfall value as 666 mm whereas in the text it indicates 1000 m. please check.
- Section 5: The description indicates that uncertainty arising from input sources. In this study, such as R, LS, K and CP are not quantified. Out of 6 variables in SE equation, four are not considered. Similarly, the text indicates that model uncertainties are considered (and immediately suggest structural uncertainty could not be quantified). In my opinion, model uncertainty is structural uncertainty (they are the same thing). So it appears that the paper is unable to take into account many variables of uncertainty assessment. In such circumstances, I don't see a good justification for the title of the paper which highlights uncertainty estimations.
- The second paragraph of section 5 Limitation is explaining very generic limitation of data and linked to uncertainty. This needs revision and I suggest not to highlight this kind of uncertainty which is there any way (such as DEM and RUSLE equations).
- In concluding remarks: various points describing findings be avoided and major conclusions can be highlighted.

**Minor comments**

- Abstract: page 1, Line 22: "Furthermore, the topographic steepness (LS) and crop
- practice (CP) factors exhibit higher uncertainties than other RUSLE factors." However, In the main text, R, LS, K and CP are not quantified for uncertainty analysis. Please check the consistency
- Page 2: Line 5-6: the 'uncertainties" is mentioned in two places, latter can be removed. (..... These uncertainties can stem from uncertainties in data)
- Please add some references for sources of uncertainty. For example, in hydrological modelling application, uncertainties are from 1) model input data 2) structural uncertainty 3) parameter uncertainty

- Page 5: Line 4: sentence not clear "role of uncertainties in input parameters on uncertainties in the estimates"
- Page 5: Line 1 indicated river as ungauged, but Line 24 suggest one gauging station. Please clarify!
- Page 6, Line 8: SE is estimated by Revised Universal Soil Loss Equation (RUSLE)......The abbreviation should be used the first time when it is mentioned. RUSLE has been mentioned many time in above sections. Same applies to others also.
- Page 6: 25. It would be useful to define the SDR with proper reference. Example: SDR is defined as the sediment yield from a catchment area divided by gross erosion of the same area.
- Figure 4a: check the legend
- Figure 7 appears before Figure 5 in the text
- Page 19: Line 8 : inconsistency in references

---

## Author Comment (AC1) · 13 Mar 2018

Dear Dr Panagos,

We sincerely thank you for your valuable comments on this manuscript. We have tried our best to incorporate all your suggestions. Detailed answers to the specific questions are given in the following paragraphs.

**Comment 1: In the current study, the erosivity factor (R-factor) has high uncertainties. I am not in favor of functions which estimate erosivity based on annual of monthly rainfall values (you can see the low quality results with large pixels in R-factor). Currently there is an increasing availability of high temporal resolution rainfall data which allow to estimate rainfall erosivity according to the principles of USLE/RUSL/e. The recent publication and data availability of Global Rainfall Erosivity has demonstrated this and there are about 250 stations with measured R-factor in India.**

**Response 1:** We agree that R factor estimated using high temporal resolution (sub – hourly) dataset based on the principles of USLE/RUSLE is better than that estimated using coarser resolution (monthly or annual) rainfall values. High-resolution rainfall datasets are available for the recent period (example R factor estimated in Global Rainfall Erosivity dataset for India uses hourly data of 250 rain gauges for 2007 – 2015). However, such high-resolution rainfall datasets are not available for the entire study period that starts from 1962 (selected as per the availability of sediment yield records). Since significant annual and decadal variability in the rainfall pattern exists over India, R factor estimated for the recent period may not be a true representative of the study period ranging from 1962 – 2008. Hence, we used IMD dataset for the study that are available at a spatial resolution of 0.25 degree for 1901 – 2013 based on more than 6,000 rain gauge stations over India. Further, the framework proposed for quantifying and propagating uncertainties in SE, SDR and SY estimates, which is the main focus of this paper, is applicable for R factor derived from different resolution datasets based on different principles.

In this study, we selected equation proposed by Babu et al. (1978) that was developed using the rainfall data from various meteorological stations in India (Eq. a in Table 2 in the manuscript). This equation is based on the linear regression between annual average rainfall amounts and R factor. Originally, this equation was proposed to estimate R factor in meter tonnes cm / ha hr unit which needs a multiplication factor of '9.8' to convert into MJ mm/ ha hr unit (Foster et al., 1981). We had missed this factor in our estimate. After revision, we have incorporated the updated value of R factor and revised the subsequent results. Revised R factor are given in Table 1 and shown in Figure 1(c).

**Comment 2: Regarding soil erodibility, the recent developments show that also soil structure and Stoniness should be taken into account. Moreover, an additional source of uncertainty has to do with interpolating methods (how did you produce surface maps from the Kfactor measurements) and the high organic carbon soils (there is literature about how to interpolate K-factor and how to face the issue of high soil organic carbon).**

**Response 2:** The present study accounts for soil structure in K factor estimates by using soil structure classification (sc) given by Wischmeier and Smith (1978). The equation (c) in Table 2 provides the equation for K factor estimate, and the structure classification codes are given in the annotation (page 15, line 23).

The Garra basin has primarily three kinds of soil textures: loam, sand and sandy loam (Figure 2c) with negligible amount of gravels. Hence, stoniness is not accounted in the K factor estimates. We will explicitly mention this fact in the revised manuscript.

Soil in the Garra basin has organic matter (OM) less than 0.2%. This value is much smaller than 4%, the maximum range of OM for which Wischmeier and Smith (1978) equation for K factor estimate is applicable.

The K factor is obtained based on NRSC (National Remote Sensing Center) soil dataset available at a spatial resolution of 56m. Thus, no interpolation is performed in estimating K factor map. Instead, the interpolation was performed by NRSC in preparing the soil data. Since NRSC did not provide the interpolation uncertainty, it was not included in the study. This limitation arising due to non-availability of interpolation error in soil data (also present in rainfall and LULC data) will be mentioned in section 5 (Limitations) of the revised manuscript.

**Comment 3: In the topographic factor, authors do not discuss the pixel size issue. There much higher uncertainty when LS-factor is calculated with pixels of 90m resolution compared too much higher resolution of 25m (all this has been discussed in European application of LS-factor). Moreover, I see values of LS-factor = 2465 .This is impossible for soil coverages.**

**Response 3:** In the literature review section, we have given references that discuss the issue of LS factor uncertainty due to cell size variation (page no. 2, line 25). However, this issue was not discussed in length because the objective of this paper is not to study the effect of cell size variation on LS factor uncertainty, but rather to provide a methodology to estimate LS factor uncertainty arising due to errors in DEM (geo-location and elevation errors). Different resolution DEMs obtained from different measurement techniques (remote sensing or ground based survey) may have different geo-location and elevation errors resulting in different LS factor uncertainties. The proposed methodology can be used to estimate LS factor uncertainty irrespective of the DEM resolution or measurement techniques used for its preparation.

Yes, LS factor cannot be so high. It was a plotting error which has been corrected. Furhter, we have improved the value of slope length exponent (m). In the previous version, the exponent is estaimted assuming rill to interrill ratio (β) as 0.67. In the revised version, β is estaimted based on basin median slope (Morgan, 2016; McCool et al., 1997), which changed m from 0.40 to 0.14. The resulting LS factor is shown in Fig 1(a) below and its range is given in Table 1.

Table 1 compares the earlier and revised estaimtes of R and LS factors, and soil erosion. Table 2 presents the revised estiamtes of sediment yield at Nanak sagar dam (NSD) and Husepur gauging station (HSG). Compared to the earlier estimates, the revised estimates are closer to observed sediment yield at both the locations.

[Figure]

Figure 1 (a) Modified LS factor (b) re-classified soil erosion map (c) Modified R factor

**Comment 4: The cover management factor is the most uncertain in USLE applications. In the manuscript it is not clear (Table 3c) how you got those C-factor ranges and how you calibrate at pixel level? The use of remote sensing on vegetation density may help you on this.**

**Response 4:** Yes, we agree that the cover management (CP) factor is the most uncertain among other RUSLE factors. This is evident in our results shown in Fig 6(c), where the magnitude of CP factor uncertainty is much higher than other factors.

The C factor ranges given in Table 3 are obtained from Morgan (2009; table 6.2, page no. 122). The ranges for different LULC classes were used to estimate uncertainty in C factor by assuming a triangular distribution that spans the entire range of C factor (equation (i) in table 2).

The vegetation density obtained using remote sensing (vegetation indices) can provide an alternative method to quantify C factor and its uncertainty. This will be mentioned in the revised manuscript.

**Comment 5: Also how did you find the P-factor values? The literature has quite different values.**

**Response 5:** The P factor is obtained from Morgan (2009; table 6.3, page no. 123). The table provides P factors for contour and strip cropping based on slope conditions.

**Comment 6: (a) The first concluding remark is not valid. this is obvious! (b) The soil erosion map could have at least 6-7 classes to show a clear distinction between low erosion , low medium , medium, high , severe, etc (with colours from Green to Red). (c) Tables should be self-explained. I don't agree with the current structure presenting the equations in the table and having the factors and annotations in separate page. It is not easy for readers.**

**Response 6:**

a) The point is not a concluding remark, but it summarises the work done. We will remove this point from the revised manuscript.
b) Figure 1(b) in this response shows the soil erosion map with six classes (increased from four in the original manuscript). We hope that the new figure distinguishes different soil erosion areas.
c) Thanks for the suggestion. In the revised manuscript, symbols will be explained along with the equations in which they are used.

Table 1 Earlier and modified R, LS and soil erosion values

| Factor | R factor (MJ mm/ha hr year) | | LS factor | | Soil Erosion (t/ha/year) | |
|---|---|---|---|---|---|---|
| | Earlier | Now | Earlier | Now | Earlier | Now |
| **Minimum** | 666.5 | 6532 | 0.09 | 0.03 | 0 | 0 |
| **Maximum** | 349.7 | 3427 | 53 | 22 | 1356 | 1423 |
| **Mean** | 467.3 | 4579.5 | 5.4 | 0.6 | 20.4 | 23 |

Table 2 Earlier and modified sediment yield values

| Station | Observed | Earlier | Now |
|---|---|---|---|
| **Nanak Sagar** | $6.4 \times 10^5$ | $8 \times 10^5$ | $6.9 \times 10^5$ |
| **Husepur** | $7.2 \times 10^6$ | $7.9 \times 10^6$ | $6.7 \times 10^6$ |

**References:**

Babu, R., Tejwani, K. G., Agarwal, M. C., & Bhushan, L. S. Distribution of erosion index and iso-erodent map of Inia. Indian Journal of Soil Conservation, 1978.

Foster, G. R., McCool, D. K., Renard, K. G., & Moldenhauer, W. C. Conversion of the universal soil loss equation to SI metric units. Journal of Soil and Water Conservation, 36(6), 355-359, 1981.

McCool, D.K., Foster, G.R. & Weesies, G.S. Slope length and steepness factors. In Renard, K.G., Foster, G.R., Weesies, G.A., et al. (eds), Predicting soil erosion by water: a guide to conservation plan- ning with the Revised Universal Soil Loss Equation (RUSLE). USDA Agricultural Handbook 703, 1997.

Morgan, R. P. C. Soil erosion and conservation. John Wiley & Sons, 2009.

Morgan, R. P. C., & Nearing, M. (Eds.). Handbook of erosion modelling. John Wiley & Sons, 2016.

Wischmeier, W. H., & Smith, D. D. Predicting rainfall erosion losses. Agricultural Handbook no. 537, US Department of Agriculture. Science and Education Administration, 1978.

---

## Author Comment (AC2) · 13 Mar 2018

**Reviewer#1**

We sincerely thank reviewer 1 for their valuable comments on this manuscript. We have tried our best to incorporate all suggestions. Detailed answers to the specific questions are given in the following paragraphs.

Comment 1: Page 2, the main objective of the manuscript is presented in lines 10-12. However, there are repeated detailed objectives below. I would suggest a more clear structure for the Introduction section.

**Response 1:** The objective present in page 2, lines 10 - 12 *"The main objective of this work is to present a methodology to quantify uncertainties in SE and SY for ungauged basins using commonly used models and easily accessible datasets."* emphasises the importance of this research and explain its usefulness to readers. Specific objectives of the study are stated in page 5, line 5 - 10. As suggested by the reviewer, we have removed lines 10-12 in page 2 and tried to avoid any repetitions in the revised manuscript.

**Comment 2: As for R in RUSLE, is it a rainfall and runoff erosivity factor according to the original model concept?**

**Response 2: Yes,** R factor is used as rainfall and runoff erosivity factor according to the original model concept proposed by Wischmeier and Smith (1978).

**Comment 3: In page 5, line 26, the discharge and sediment load records for 16 years are available at the stations. The manuscript should give more details on model calibration and validation.**

**Response 3:** Sediment load records for Husepur gauging station (HSG) and Nanak sagar dam (NSD) are available for 16 and 40 years, respectively. RUSLE model was originally proposed to apply for long term (> 20 years) soil erosion estimation (Wischmeier, 1959; Nearing et al., 2017). Hence we have validated our model with average sediment load at NSD and HSG during the observation period. The results are presented in Figure 8, page 31.

**Comment 4: There is a large reservoir built in 1962, which may play an imporant role in sediment trapping. The sedimentation rate data can be used for sediment yield calibration. The SDR should also consider the effect of reservoir trapping, though the model is empirical.**

**Response 4:** Yes, Nanak Sagar reservoir has a significant effect on sediment trapping (6.4 x 105 tones/year). We have incorporated the effect of reservoir on soil erosionand sediment yield estimation by using the methodlogy proposed by Sharda and Ojasvi, 2016. Gross soil erosion for the Garra basin is estimated by extracting the area covered by the reservoir. It is called as gross soil erosion for free basin area (Total basin area – reservoir basin area). The gross soil erosion for free basin area and SDR are used to estimate sediment yield at Husepur.

**Comment 5: When compared the annual rainfall and rainfall erosivity, I found the R factor is much lower than the regions with similar rainfall amount, I doubt the proposed the method for R estimation. As well, the very coarse rainfall data might be the dominant factor influencing the simulation results, rather then the R factor itself.**

**Response 5:** In this study, we selected equation proposed by Babu et al. (1978) that was developed using the rainfall data from various meteorological stations in India (Eq. a in Table 2 in the manuscript). This equation is based on the linear regression between annual average rainfall amounts and R factor. Originally, this equation was proposed to estimate R factor in meter tonnes cm / ha hr unit which needs a multiplication factor of '9.8' to convert into MJ mm/ ha hr unit (Foster

et al., 1981). We had missed this factor in our estimate. After revision, we have incorporated the updated value of R factor and revised the subsequent results. Revised R factor are given in Table 1 and shown in Figure 1(c).

Please also see our response to comment 7.

**Comment 6: Soil map is rough too, I would suggest to do a field survey for sampling, or obtain a relative detailed soil data.**

**Response 6:** We have used NRSC soil data (1:50,000; 25 m), which is described in Table 1, page 23. This is the best soil dataset available for this region in terms of spatial resolution and is based on field surveys. Obtaining new field survey data for the study area will be repetitive, cumbersome and outside the scope of this study. We have re-classified the soil map into soil textural classes, namely loam, sand and sandy loam (Figure 2 (c), page 26) which is why it looks rough. On the other hand, this dataset has 11 soil classes (shown in Figure 1(b) below), and for each class, K factor is estimated.

Comment 7: As for LS factor, the maximum value is around 2500, this is extremely high due to the high gradients. This means the LS factor may be overestimated for the steep area, since the RUSLE model was originally developed for estimating soil erosion in relative gentle arable land.

**Response 7:** Yes, LS factor cannot be so high. It was a plotting error which has been corrected. Furhter, we have improved the value of slope length exponent (m). In the previous version, the exponent is estaimted assuming rill to interrill ratio ( $\beta$ ) as 0.67. In the revised version,  $\beta$  is estaimted based on basin median slope (Morgan, 2016; McCool et al., 1997), which changed m from 0.40 to 0.14. The resulting LS factor is shown in Fig 1(a) below and its range is given in Table 1.

Table 1 compares the earlier and revised estaintes of R and LS factors, and soil erosion. Table 2 presents the revised estiantes of sediment yield at Nanak sagar dam (NSD) and Husepur gauging station (HSG). Compared to the earlier estimates, the revised estimates are closer to observed sediment yield at both the locations.

**Comment 8: When I saw the data listed in Table 2, the resolution for different data may cause high uncertainties for modeling results. The resolution of the spatial data highly influence the data quality, such as LS factor, K-factor, C and P factor.**

**Response 8:** Yes, due to variation of resolution of different data may produce uncertainty in the modelled results. In this work, we have tried to estimate and propagate the uncertainty using easily available and most commonly used dataset for the Indian region in soil erosion and sediment yield prediction. It is explained in "Limitation" section, page 13.

Figure 1 (a) Modified LS factor (b) Soil class map (c) Rainfall erosivity

|         | R factor (MJ mm/ha
hr year) |         | LS factor |         | Soil Erosion
(t/ha/year) |         |
|---------|--------------------------------|---------|-----------|---------|-----------------------------|---------|
| Factor  | Earlier                        | Revised | Earlier   | Revised | Earlier                     | Revised |
| Minimum | 666.5                          | 6532    | 0.09      | 0.03    | 0                           | 0       |
| Maximum | 349.7                          | 3427    | 53        | 22      | 1356                        | 1423    |
| Mean    | 467.3                          | 4579.5  | 5.4       | 0.6     | 20.4                        | 23      |

**Table 1 Earlier and modified R, LS and soil erosion values**

Table 2 Earlier and modified sediment yield values (units)

| Station     | Observed              | Earlier               | Revised               |
|-------------|-----------------------|-----------------------|-----------------------|
|             | 6.4 X 10 5 | 8 X 10⁵               | 6.9 X 10⁵             |
| Nanak Sagar |                       |                       |                       |
|             | 7.2 X 10 6 | 7.9 X 10 6 | 6.7 X 10 6 |
| Husepur     |                       |                       |                       |

**References:**

Babu, R., Tejwani, K. G., Agarwal, M. C., & Bhushan, L. S. Distribution of erosion index and isoerodent map of Inia. Indian Journal of Soil Conservation, 1978.

Foster, G. R., McCool, D. K., Renard, K. G., & Moldenhauer, W. C. Conversion of the universal soil loss equation to SI metric units. Journal of Soil and Water Conservation, 36(6), 355-359, 1981.

McCool, D.K., Foster, G.R. & Weesies, G.S. Slope length and steepness factors. In Renard, K.G., Foster, G.R., Weesies, G.A., et al. (eds), Predicting soil erosion by water: a guide to conservation plan- ning with the Revised Universal Soil Loss Equation (RUSLE). USDA Agricultural Handbook 703, 1997.

Morgan, R. P. C., & Nearing, M. (Eds.). Handbook of erosion modelling. John Wiley & Sons, 2016.

Nearing, M. A., Yin, S. Q., Borrelli, P., & Polyakov, V. O. Rainfall erosivity: An historical review. Catena, 157, 357-362, 2017.

Sharda, V. N., & Ojasvi, P. R. A revised soil erosion budget for India: role of reservoir sedimentation and land-use protection measures. Earth Surface Processes and Landforms, 41(14), 2007-2023, 2016.

USDA. Sediment sources, yields, and delivery ratios. National Engineering Handbook, Section 3 Sedimentation, 1972.

Wischmeier, W.H. A rainfall erosion index for a universal soil loss equation. Soil Sci. Soc. Am. Proc. 23, 322–326, 1959.

Wischmeier, W. H., & Smith, D. D. Predicting rainfall erosion losses. Agricultural Handbook no. 537, US Department of Agriculture. Science and Education Administration, 1978.

---

## Author Comment (AC3) · 13 Mar 2018

**Reviewer#2**

We sincerely thank reviewer 1 for their valuable comments on this manuscript. We have tried our best to incorporate all suggestions. Detailed answers to the specific questions are given in the following paragraphs.

Comment 1: Page 6: Line 20-24: How the spatial variability of rainfall is taken into account. Since the measured rainfall data is from stations, which interpolation method was adopted to spatially distribution the rainfall data for each grid.

**Response:** We have used gridded rainfall dataset developed by India Meteorological Department (Rajeevan and Bhate, 2009). The dataset is available at 0.25 degree resolution and is obtained by interpolating more than 4,000 measured rainfall stations data using Shepard's interpolation method (Shepard, 1968).

Comment 2: Section 3.4: Step by Step procedure: is not a methodological step. Probably the sentence can just continue without the section.

**Response:** As suggested, we have removed this section heading in the revised version of the manuscript.

Comment 3: 4.1.1: Figure 4(a) suggest the highest rainfall value as 666 mm whereas in the text it indicates 1000 m. please check.

**Response:** Figure 4(a) is the rainfall erosivity map which is estimated by using average annual rainfall values. The values are reported in MJ mm/ha hr y units. In the text (page 10, line 25), we have reported that in upper mountainous region of the study area the average annual rainfall exceeds 1000 mm. To avoid any confusion, we have now prominently shown the units of R factor in Figure 4(a).

Comment 4: Section 5: The description indicates that uncertainty arising from input sources. In this study, such as R, LS, K and CP are not quantified. Out of 6 variables in SE equation, four are not considered. Similarly, the text indicates that model uncertainties are considered (and immediately suggest structural uncertainty could not be quantified). In my opinion, model uncertainty is structural uncertainty (they are the same thing). So, it appears that the paper is unable to take into account many variables of uncertainty assessment. In such circumstances, I don't see a good justification for the title of the paper which highlights uncertainty estimations.

**Response:** We have quantified uncertainty for all six factor of RUSLE. These uncertainties can stem from data (measurement errors, coarse spatial and temporal resolution, missing values), model (parameter uncertainty, structural uncertainty, algorithmic or numerical uncertainty), and stochastic nature of the soil erosion process. Structural uncertainty is part of model uncertainty (Beven and Brazier, 2011). However, in this study, we have used easily available uncertainty for different factors (Page 2, Line 9 - 11; Page 13, Line 4 - 7).

Comment 5: The second paragraph of section 5 Limitation is explaining very generic limitation of data and linked to uncertainty. This needs revision and I suggest not to highlight this kind of uncertainty which is there any way (such as DEM and RUSLE equations).

**Response:** Agreed. We have removed the concerned lines from section 5 in the revised manuscript.

Comment 6: In concluding remarks: various points describing findings be avoided and major conclusions can be highlighted.

**Response:** We have renamed section 6 as 'Summary and Concluding Remarks' and have highlighted the major conclusions from this study.

Comment 7: Abstract: page 1, Line 22: "Furthermore, the topographic steepness (LS) and crop practice (CP) factors exhibit higher uncertainties than other RUSLE factors." However, In the main text, R, LS, K and CP are not quantified for uncertainty analysis. Please check the consistency.

**Response:** Uncertainty assessment has been done for all six factors of RUSLE equation (Please see response to comment 4). Thereafter, uncertainty comparisons suggest that LS and CP factor exhibit higher uncertainty.

Comment 8: Page 2: Line 5-6: the 'uncertainties" is mentioned in two places, latter can be removed. (..... These uncertainties can stem from uncertainties in data).

Response: Done.

Comment 9: Please add some references for sources of uncertainty. For example, in hydrological modelling application, uncertainties are from 1) model input data 2) structural uncertainty 3) parameter uncertainty

**Response:** We have cited the literature published by Beven and Brazier (2011) and JCGM (2008) on source of uncertainties in the erosion model predictions (Page 2, Line 4). However, research on uncertainties estimation for soil erosion and sediment yield is very few which are mostly covered in the Introduction section (Page 2, Line 20 – Page 3, Line 33).

**Comment 10: Page 5: Line 4: sentence not clear "role of uncertainties in input parameters on uncertainties in the estimates"**

**Response:** We have corrected the sentence in the revised manuscript. The modified sentence is "The Garra River, a Himalayan tributary of River Ganga, was selected for demonstration of the developed methodology and for investigating the role of uncertainties in input parameters and SE and SY estimates".

**Comment 11: Page 5: Line 1 indicated river as ungauged, but Line 24 suggest one gauging station. Please clarify!**

**Response:** The study basin has two gauging stations where sediment yield is measured - (1) Nanak Sagar Dam at upstream and (2) Husepur at downstream. Validation of modelled results are done by using data from these two gauge stations. However, Line 1 indicates that presented methodology can be applied to an ungauged basin.

Comment 12: Page 6, Line 8: SE is estimated by Revised Universal Soil Loss Equation (RUSLE).....The abbreviation should be used the first time when it is mentioned. RUSLE has been mentioned many time in above sections. Same applies to others also.

**Response:** Agreed and done.

Comment 13: Page 6: 25. It would be useful to define the SDR with proper reference. Example: SDR is defined as the sediment yield from a catchment area divided by gross erosion of the same area.

**Answer:** We have defined SDR in Page 4 lines 8 – 10 and provided reference for it (Walling, 1983; Richards, 1993; USDA, 1972; De Vente et al., 2007).

**Comment 14: Figure 4a: check the legend**

**Response:** Figures 4(a) is a rainfall erosivity map which is derived from average annual rainfall values.

**Comment 15: Figure 7 appears before Figure 5 in the text**

Response: Corrected.

**Comment 16: Page 19: Line 8: inconsistency in references**

**Response:** Checked and corrected.

**References:**

Beven, K. J., & Brazier, R. E. Dealing with uncertainty in erosion model predictions. Handbook of Erosion Modelling, 52-79, 2011.

De Vente, J., Poesen, J., Arabkhedri, M., & Verstraeten, G. The sediment delivery problem revisited. Progress in Physical Geography, 31(2), 155-178, 2007.

JCGM. Evaluation of measurement data - Guide to the expression of uncertainty in measurement. Working Group 1 of the Joint Committee for Guides in Metrology (JCGM/WG 1), 2008.

Rajeevan, M., & Bhate, J. A high resolution daily gridded rainfall dataset (1971–2005) for mesoscale meteorological studies. Current Science, 96(4), 558-562, 2009.

Richards, K. Sediment delivery and the drainage network. Channel network hydrology, 221-254, 1993.

Shepard, D., A two-dimensional interpolation function for irregularly-spaced data. In *Proceedings* of the 1968 23rd ACM national conference (pp. 517-524). ACM, 1968.

USDA. Sediment sources, yields, and delivery ratios. National Engineering Handbook, Section 3 Sedimentation, 1972.

Walling, D. E. The sediment delivery problem. Journal of hydrology, 65(1), 209-237, 1983.